# UBE3A-mediated p18/LAMTOR1 ubiquitination and degradation regulate mTORC1 activity and synaptic plasticity

Jiandong Sun[1], Yan Liu[2], Yousheng Jia[3], Xiaoning Hao[1], Wei ju Lin[1], Jennifer Tran[1], Gary Lynch[3], Michel Baudry[2], Xiaoning Bi[1]*

[1]Department of Basic Medical Sciences, College of Osteopathic Medicine of the Pacific, Western University of Health Sciences, Pomona, United States; [2]Graduate College of Biomedical Sciences, Western University of Health Sciences, Pomona, United States; [3]Department of Psychiatry, University of California, Irvine, United States

**Abstract** Accumulating evidence indicates that the lysosomal Ragulator complex is essential for full activation of the mechanistic target of rapamycin complex 1 (mTORC1). Abnormal mTORC1 activation has been implicated in several developmental neurological disorders, including Angelman syndrome (AS), which is caused by maternal deficiency of the ubiquitin E3 ligase UBE3A. Here we report that Ube3a regulates mTORC1 signaling by targeting p18, a subunit of the Ragulator. Ube3a ubiquinates p18, resulting in its proteasomal degradation, and Ube3a deficiency in the hippocampus of AS mice induces increased lysosomal localization of p18 and other members of the Ragulator-Rag complex, and increased mTORC1 activity. p18 knockdown in hippocampal CA1 neurons of AS mice reduces elevated mTORC1 activity and improves dendritic spine maturation, long-term potentiation (LTP), as well as learning performance. Our results indicate that Ube3a-mediated regulation of p18 and subsequent mTORC1 signaling is critical for typical synaptic plasticity, dendritic spine development, and learning and memory.

DOI: https://doi.org/10.7554/eLife.37993.001

*For correspondence:
xbi@westernu.edu

## Introduction

The mechanistic target of rapamycin (mTOR) is a highly conserved and ubiquitously expressed protein kinase complex, which plays important roles in cell survival, growth, and metabolism. mTOR, consists of two complexes, mTORC1 and mTORC2, which integrate extracellular signals (growth factors, neurotransmitters, nutrients, etc.) with intracellular energy levels and cellular stress status to regulate many important cellular functions (*Laplante and Sabatini, 2012*; *Takei and Nawa, 2014*). Not surprisingly, abnormal mTOR signaling has been implicated in various neurodevelopmental disorders and neuropsychiatric and neurological diseases (*Costa-Mattioli and Monteggia, 2013*). Recent evidence indicates that amino acid-induced lysosomal recruitment of mTORC1 is essential for its full activation (*Jewell et al., 2013*). In the presence of amino acids, mTORC1 is activated by binding to heterodimers consisting of Rag small guanosine triphosphatases (GTPases) RagA/B in their GTP-bound status, and RagC/D in their GDP-bound status (*Ham et al., 2016*). Lysosomal localization of Rag dimers is maintained through their binding to the Ragulator complex, which consists of p18 (also known as LAMTOR1), p14 (LAMTOR2), MP1 (LAMTOR3), C7orf59 (LAMTOR4), and HBXIP (LAMTOR5) proteins (*Bar-Peled et al., 2012*; *Nada et al., 2009*; *Sancak et al., 2010*); acylation of p18 is essential for anchoring the Ragulator complex to endosomal/lysosomal membranes (*Nada et al., 2009*). Although it has been shown that lysosomal localization and interaction with Rag GTPases are essential for p18 to regulate mTORC1 activation (*Sancak et al., 2010*), little is known

regarding the regulation of p18 levels. A previous study combining single-step immuno-enrichment of ubiquitinated peptides with high-resolution mass spectrometry revealed that p18 was ubiquitinated at residues K20 and K31 in HEK cells and MV4-11 cells (*Wagner et al., 2011*). However, the E3 ligase responsible for p18 ubiquitination was not identified. Furthermore, it is not known whether the Ragulator-Rag complex regulates mTORC1 in the central nervous system (CNS) in a way similar to that in peripheral tissues.

UBE3A, an E3 ligase in the ubiquitin-proteasomal system, plays important roles in brain development and normal function, as UBE3A deficiency results in Angelman syndrome (AS) (*Williams et al., 1990*), while UBE3A over-expression increases the risk for autism (*Cook et al., 1997*). We recently reported that imbalanced signaling of the mTOR pathway, with increased mTORC1 and decreased mTORC2 activation, can result in motor dysfunction and abnormal dendritic spine morphology of Purkinje neurons in AS mice (*Sun et al., 2015a*). Similar abnormal mTOR signaling is critically involved in Ube3a deficiency-induced impairment in hippocampal synaptic plasticity and fear-conditioning memory (*Sun et al., 2016*). Furthermore, inhibition of mTORC1 by rapamycin treatment not only reduced mTORC1 activity but also normalized mTORC2 activity, suggesting that mTORC1 over-activation is the trigger for alterations in mTOR signaling in AS mice. However, it remains unknown how Ube3a deficiency results in mTORC1 over-activation. The present study investigated the potential regulation of p18 levels by Ube3a. We demonstrate that Ube3a directly ubiquitinates p18 and targets it for proteasomal degradation, which normally limits mTORC1 signaling and activity-dependent synaptic remodeling. In the absence of Ube3a, p18 accumulates in neurons, resulting in mTORC1 over-activation, abnormal synaptic morphology, and impaired synaptic plasticity and learning. These findings reveal a previously unidentified regulatory mechanism for mTORC1 activation and suggest potential therapeutic targets for cognitive disorders associated with abnormal mTORC1 signaling.

## Results

### p18 is a Ube3a substrate

Although it has been shown that p18 plays essential roles in mTOR and MAP kinase signaling and other cellular functions, very little is known regarding its biosynthesis and degradation. We first determined whether Ube3a could regulate p18 levels in heterologous cells. Western blot analysis showed that Ube3a knockdown (KD) in COS-1 cells by siRNA resulted in increased p18 levels, as compared with scrambled control siRNA (*Figure 1A*). Sequence analysis revealed the presence of six lysine residues in p18, which could represent ubiquitination sites (*Figure 1B*). To assess whether p18 could be a Ube3a substrate, we first determined whether these two proteins exhibited direct interactions. Co-immunoprecipitation experiments using extracts from COS-1 cells transfected with Ube3a and Flag-p18 showed that p18 could bind to Ube3a in an E3 ligase activity-independent manner, as p18 could also bind to an inactive form of Ube3a with a mutation in its catalytic site, Ube3a-C833A (*Kumar et al., 1999*) (referred to as ΔUbe3a hereafter) (*Figure 1C*). In vitro ubiquitination assays using purified recombinant p18 and a Ube3a ubiquitination assay kit showed that p18 ubiquitination was only observed in the presence of Ube3a, ubiquitin, E1, E2, and ATP (*Figure 1D*). We then determined whether Ube3a could ubiquitinate p18 in intact cells using His-ubiquitin pull-down assay. COS-1 cells were co-transfected with p18 and His-ubiquitin plus an empty vector, or Ube3a or ΔUbe3a. Ubiquitinated proteins were extracted by $Co^{2+}$-affinity chromatography and analyzed by Western blot. Co-transfection with Ube3a, but not ΔUbe3a, resulted in massive p18 ubiquitination (*Figure 1E* and *Figure 1—figure supplement 1A*). In addition, transfection with Ube3a, but not ΔUbe3a, resulted in decreased p18 levels (*Figure 1F* and *Figure 1—figure supplement 1B*), indicating that Ube3a-mediated regulation of p18 levels depends on its E3 ligase activity and p18 ubiquitination. Finally, to confirm that Ube3a-mediated p18 ubiquitination was taking place on lysine residues, we simultaneously mutated all lysine residues into arginine (ΔK) in Flag-p18. COS-1 cells were first transfected with either Ube3a siRNA or a scrambled control siRNA, followed by transfection with Flag-p18 or Flag-p18ΔK and His-ubiquitin. His-ubiquitin pull-down assay showed that levels of p18-immunopositive high molecular weight bands, that is, ubiquitinated p18, were significantly reduced following Ube3a siRNA KD (*Figure 1G* and *Figure 1—figure supplement 1C*), and the degree of reduction (100.0 ± 1.8 vs. 66.8 ± 3.7, n = 4, p<0.001) corresponded to the extent of

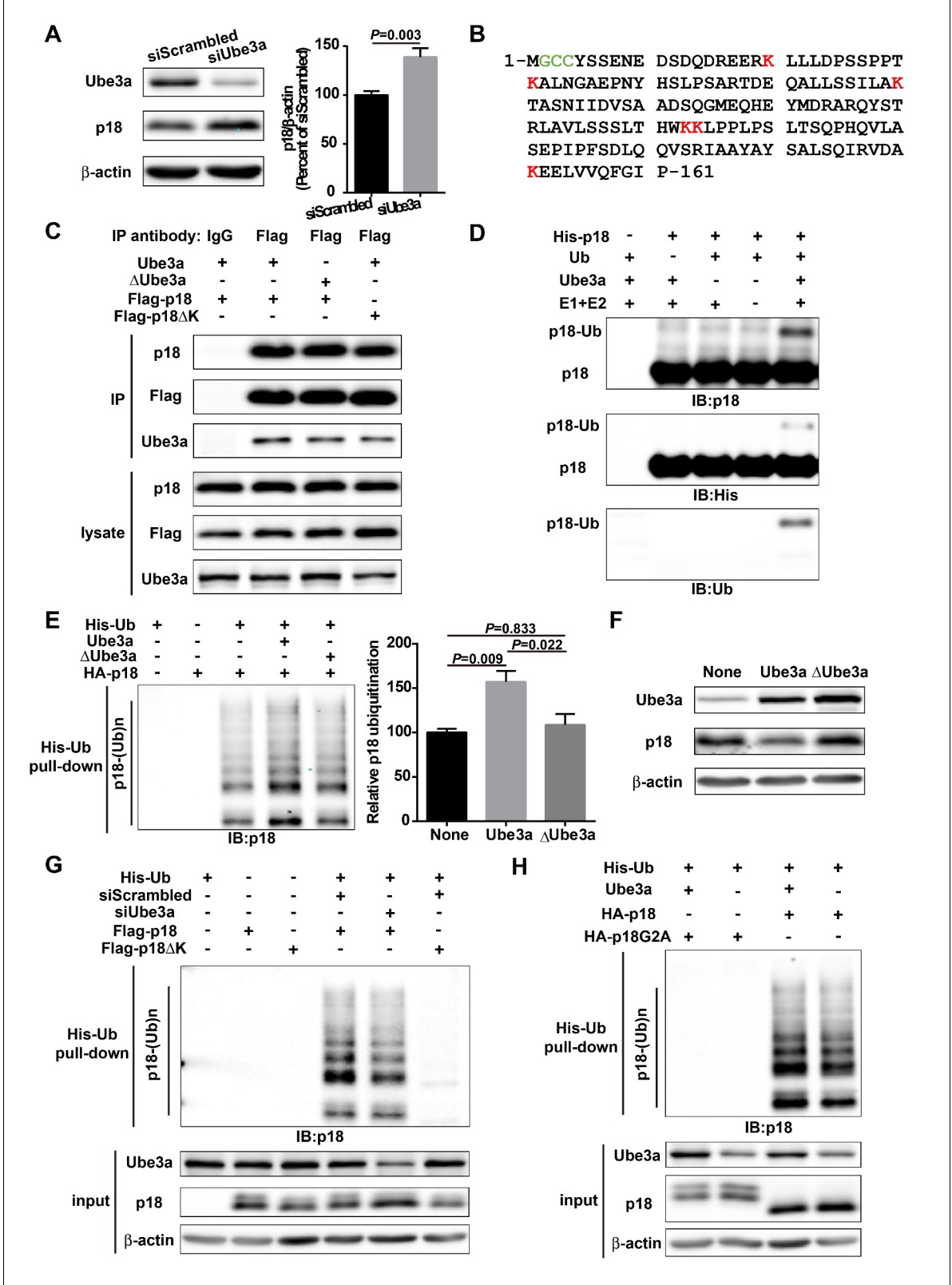

**Figure 1.** p18 is a Ube3a substrate. (**A**) Western blot analysis using anti-Ube3a, p18, or β-actin antibodies of lysates from COS-1 cells transfected with scrambled siRNA or Ube3a siRNA. Right, quantitative analysis of blots. N = 6 independent experiments, p=0.003 (unpaired, two-tailed Student's t-test). (**B**) Amino acid sequence of human p18. G2 is a myristoylation site. C3 and C4 are palmitoylation sites. K20, K31, K60, K103, K104, and K151 are potential ubiquitination sites. (**C**) Interaction between p18 and Ube3a. Lysates from COS-1 cells transfected with the indicated cDNAs in expression

*Figure 1 continued on next page*

*Figure 1 continued*

vectors were immunoprecipitated with an anti-Flag antibody or control IgG and probed with the indicated antibodies. The presence of Flag-p18 in precipitates was confirmed with anti-p18 and anti-Flag antibodies. (D) In vitro ubiquitination of p18 by recombinant Ube3a. Reaction products were analyzed by Western blots with p18, His, and ubiquitin antibodies. Note that the p18-Ub band is present only when all reaction elements are added. (E) Over-expression of Ube3a, but not ΔUbe3a, enhances p18 ubiquitination in COS-1 cells. His-tagged ubiquitinated proteins in cells co-transfected with HA-p18 plus empty vectors (None, but with endogenous Ube3a), wild-type Ube3a (Ube3a), or its inactive form Ube3a-C833A (ΔUbe3a) were precipitated using Talon resin and probed with anti-p18 antibodies. Ubiquitinated p18 proteins are labeled with 'p18-(Ub)n'. Right, quantification of the relative abundance of ubiquitinated p18 (means ± SEM, p=0.009 None vs. Ube3a, p=0.022 Ube3a vs. ΔUbe3a, p=0.833 None vs. ΔUbe3a, n = 3 independent experiments, one-way ANOVA with Tukey's post hoc analysis). (F) Western blot analysis using anti-Ube3a, p18, or β-actin antibodies on lysates from COS-1 cells transfected with empty vector, Ube3a, or ΔUbe3a vectors. (G) siRNA knockdown of Ube3a in COS-1 cells reduces p18 ubiquitination. COS-1 cells were incubated with Ube3a siRNA or scrambled control siRNA 48 hr before transfection with Flag-p18 or Flag-p18ΔK and His-ubiquitin. Twenty-four hours later, ubiquitinated proteins were isolated by $Co^{2+}$-affinity chromatography. Levels of ubiquitinated p18 protein (p18-(Ub)n, upper panel) were determined by Western blots. Levels of input proteins were also evaluated by Western blots probed with Ube3a, p18, and β-actin antibodies (lower panel). (H) His-ubiquitin pull-down assay performed using HA-p18 or HA-p18G2A. Upon purification, levels of ubiquitinated p18 (upper panel) were determined by Western blot analysis. Lower panel, input of Ube3a, p18, and β-actin. See also *Figure 1—figure supplement 1* and *Figure 1—source data 1*.

DOI: https://doi.org/10.7554/eLife.37993.002

The following source data and figure supplement are available for figure 1:

**Source data 1.** Quantitative analyses of Western blots used for *Figure 1* and *Figure 1—figure supplement 1*.
DOI: https://doi.org/10.7554/eLife.37993.004
**Figure supplement 1.** His-ubiquitin pull-down analyses of p18 ubiquitination.
DOI: https://doi.org/10.7554/eLife.37993.003

Ube3a down-regulation (100.0 ± 2.4 vs. 60.9 ± 5.2, n = 4, p<0.001). Furthermore, p18 ubiquitination was abolished by K-R mutations (*Figure 1G* and *Figure 1—figure supplement 1C*). ΔK mutations did not significantly affect the interaction between p18 and Ube3a (*Figure 1C*). These results confirmed that p18 is ubiquitinated by Ube3a at lysine residues. To further clarify which lysine residue(s) of p18 might be ubiquitinated, we engineered proteins with individual lysine to arginine substitution (K20R, K31R, K60R, K103/104R, and K151R). His-ubiquitin pull-down assays indicated that K20R, K31R, and K151R had little effect on p18 ubiquitination levels (*Figure 1—figure supplement 1G*). However, K60R and K103/104R greatly reduced p18 ubiquitination (*Figure 1—figure supplement 1G*). These results suggest that residues K60, K103, and K104 are more likely to be ubiquitinated.

Previous studies have revealed that p18 is anchored to lysosomal membranes through myristate and palmitate modifications at G2 and C3/C4, respectively (*Nada et al., 2014*). We confirmed that wild-type (WT) p18 was indeed localized at the lysosomal surface, while its myristoylation-defective mutant, p18G2A, failed to localize to the lysosomal surface and partially (because of the existence of endogenous p18) blocked mTORC1 activation (*Figure 1—figure supplement 1D–E*), and nearly completely lost its ability to be ubiquitinated (*Figure 1H* and *Figure 1—figure supplement 1F*). These results suggest that myristoylation-dependent lysosomal localization of p18 is critical for both mTORC1 activation and Ube3a-mediated p18 ubiquitination.

## p18 is essential for lysosomal localization of Ragulator and RagGTPases in hippocampal neurons

As there is little information regarding p18 in the CNS, we next characterized p18 expression in hippocampal neurons. Double immunolabeling with antibodies against p18 and LAMP2, a well-characterized lysosomal marker (*Eskelinen, 2006*), showed that p18 was co-localized with LAMP2, not only in cell bodies but also in dendrites of cultured mouse hippocampal neurons (*Figure 2A*). To test whether p18 could interact with other members of the Ragulator complex in neurons as in other cell types, neuronal lysates were immunoprecipitated from cultured hippocampal neurons with a p18 antibody and the precipitated material was probed with anti-p18, anti-p14, or anti-MP1 antibodies. p18, p14, and MP1 were detected in anti-p18- but not in control IgG-pull-down proteins (*Figure 2B*). Immunoprecipitates prepared from hippocampal lysates of WT mice with a RagA antibody, but not a control IgG, consistently contained RagA, RagB, RagC, p18, and p14 (*Figure 2B*). These results indicate that p18 interacts with other members of the Ragulator complex and that the Ragulator interacts with Rag GTPases in hippocampal neurons, as in other cell types.

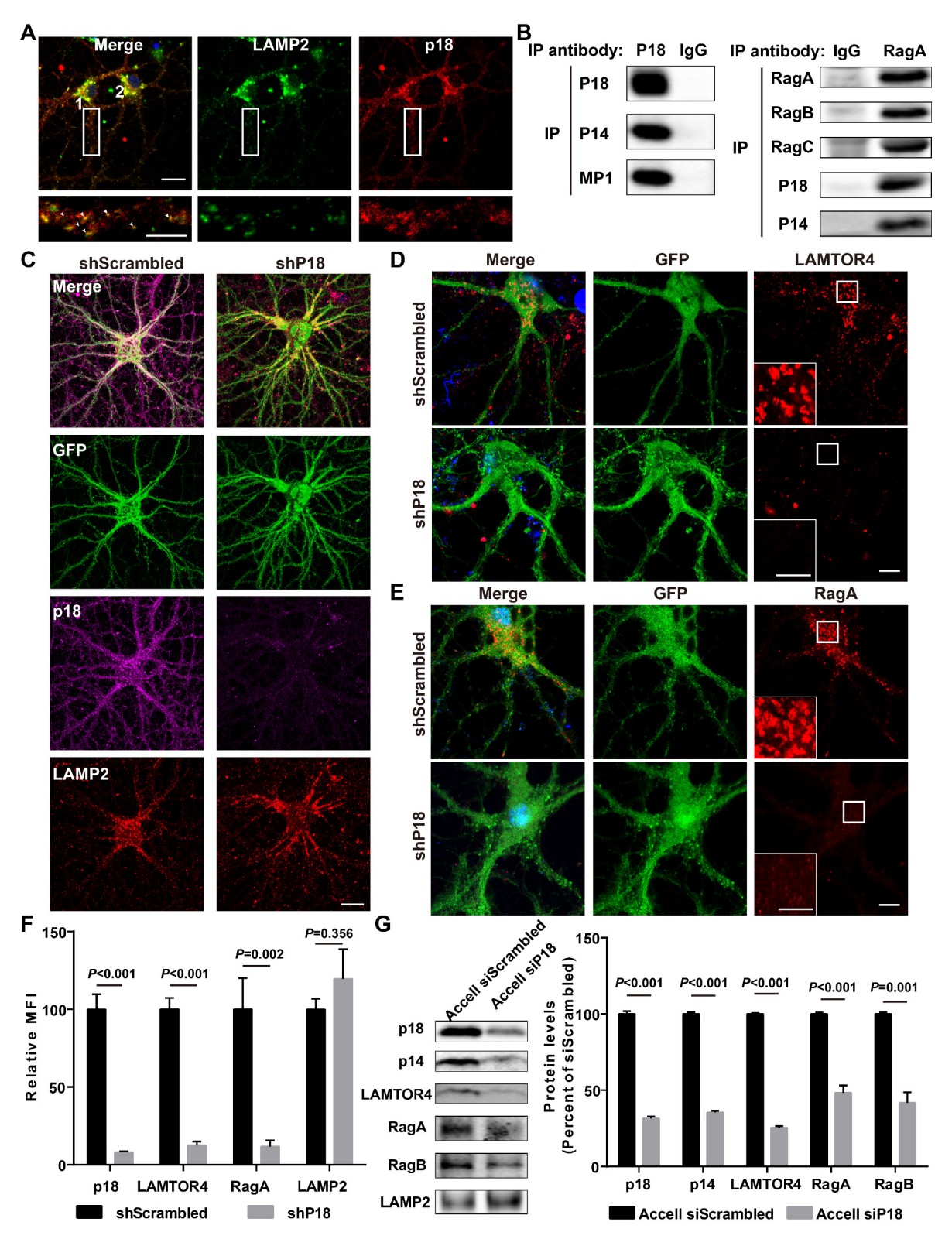

**Figure 2.** Characterization of p18 in hippocampal neurons. (**A**) Images of cultured hippocampal neurons co-immunostained for lysosomal protein LAMP2 (green) and p18 (red). Insets are enlarged images of LAMP2- and p18-immunoreactive puncta along the dendrites. Arrowheads indicate co-localized puncta. Scale bar: top, 20 µm; inset, 10 µm. (**B**) Left, p18 forms a complex with p14 and MP1 in hippocampal neurons. Lysates from cultured hippocampal neurons were immunoprecipitated with an anti-p18 antibody or control IgG and probed with the indicated antibodies. Right, RagA co-

*Figure 2 continued on next page*

*Figure 2 continued*

immunoprecipitates RagB, RagC, p18, and p14. Lysates from mouse hippocampi were immunoprecipitated with an anti-RagA antibody or control IgG and probed with the indicated antibodies. (C) Images of cultured hippocampal neurons co-immunostained for p18 (magenta) and LAMP2 (red). Neurons were infected with shRNA AAV directed against p18 with GFP co-expression or scrambled shRNA control before processing for immunofluorescence assay and imaging. Scale bar, 20 µm. (D) Images of hippocampal neurons stained for LAMTOR4 (red). Cells were infected and processed as in (C). Scale bar, 10 µm; inset, 5 µm. (E) Images of hippocampal neurons stained for RagA (red). Cells were infected and processed as in (C). Scale bar, 10 µm; inset, 5 µm. (F) Quantification of fluorescent signals for p18 (n = 13, p<0.001), LAMTOR4 (n = 11, p<0.001), RagA (n = 6, p=0.002), and LAMP2 (n = 6, p=0.356) in control shRNA and p18 shRNA-infected neurons shown in C– E. Student's t-test. Note that n refers to the number of culture dishes analyzed. (G) Left, Western blot analysis of p18, p14, LAMTOR4, RagA, RagB, and LAMP2 in enriched lysosomal fractions prepared from WT neurons transfected with Accell control or p18 siRNA. Right, quantitative analysis of blots. Results are expressed as % of values in control siRNA-transfected WT neurons and shown as means ± SEM N = 3 independent experiments, p<0.001 for p18, p14, LAMTOR4, and RagA, p=0.001 for RagB (unpaired, two-tailed Student's t-test). See also *Figure 2—figure supplement 1* and *Figure 2—source data 1*.

DOI: https://doi.org/10.7554/eLife.37993.005

The following source data and figure supplement are available for figure 2:

**Source data 1.** Quantitative analyses of images and Western blots used for *Figure 2* and *Figure 2—figure supplement 1*.
DOI: https://doi.org/10.7554/eLife.37993.007
**Figure supplement 1.** Localization of LAMTOR4 and RagA in hippocampal neurons, and representation of organelle marker proteins in individual OptiPrep™ fractions.
DOI: https://doi.org/10.7554/eLife.37993.006

To further investigate whether p18 also serves as an anchor for the Ragulator-Rag complex in the brain, neurons were infected with shRNA AAV directed against p18 to decrease p18 expression and with GFP co-expression to visualize infected neurons, and the effects on the Ragulator-Rag complex were determined. Confocal images of infected neurons indicated that p18 shRNA infection efficiently reduced p18 expression in cultured neurons, while lysosomal morphology was not obviously affected (*Figure 2C and F*). Notably, levels of lysosome-localized LAMTOR4 and RagA (*Figure 2— figure supplement 1A,B*) were significantly reduced following p18 shRNA KD in neurons (*Figure 2D–F*). Lysosomal fractions were also prepared from neuronal cultures treated with Accell control or p18 siRNA to determine the levels of members of the Ragulator-Rag complex. Western blot analysis demonstrated that the lysosomal fraction was enriched with the lysosomal protease cathepsin B, but did not contain the mitochondrial marker COXIV (*Figure 2—figure supplement 1C*). Consistent with the immunofluorescence results, lysosomal levels of p18, p14, LAMTOR4, as well as RagA and RagB were significantly reduced by p18 KD (*Figure 2G*). Thus, p18 is required for lysosomal targeting of the Ragulator-Rag complex in neurons.

## Ube3a regulates p18 levels in a proteasome-dependent manner in hippocampal neurons

We then determined whether Ube3a deficiency in neurons could result in increased p18 levels using AS mice. Western blot results showed that p18 levels were markedly increased in crude membrane fractions (P2) of hippocampus from AS mice compared with WT mice (*Figure 3A* and *Figure 3—figure supplement 1A*), while there was no significant change in levels of p14, MP1, as well as Rag GTPases (*Figure 3A* and *Figure 3—figure supplement 1A*). Importantly, although rapamycin treatment of AS mice normalized mTORC1 and mTORC2 signaling (*Sun et al., 2016*), it did not reverse the increase in p18 levels in AS mice (*Figure 3—figure supplement 1A*, same samples as those used in *Sun et al., 2016*). These results suggest that increased p18 levels in AS mice are independent of mTORC1 activity, and that increased mTORC1 activity might be downstream of p18 level increases.

Our data suggested that increased p18 levels in hippocampus of AS mice could result from the lack of Ube3a-mediated p18 ubiquitination and subsequent degradation. To confirm this possibility, we first showed that p18 was co-immunoprecipitated with Ube3a in cultured hippocampal neurons from WT mice (*Figure 3B*). Ubiquitinated proteins from cultured hippocampal neurons or P2 fractions of hippocampi from WT and AS mice were immunoprecipitated with ubiquitin antibodies under denaturing conditions, and precipitated proteins were processed for Western blot with ubiquitin and p18 antibodies. Both p18 and ubiquitin antibodies labeled high molecular weight bands, and the intensity of p18-immunopositive bands was much weaker in samples from AS mice than WT mice

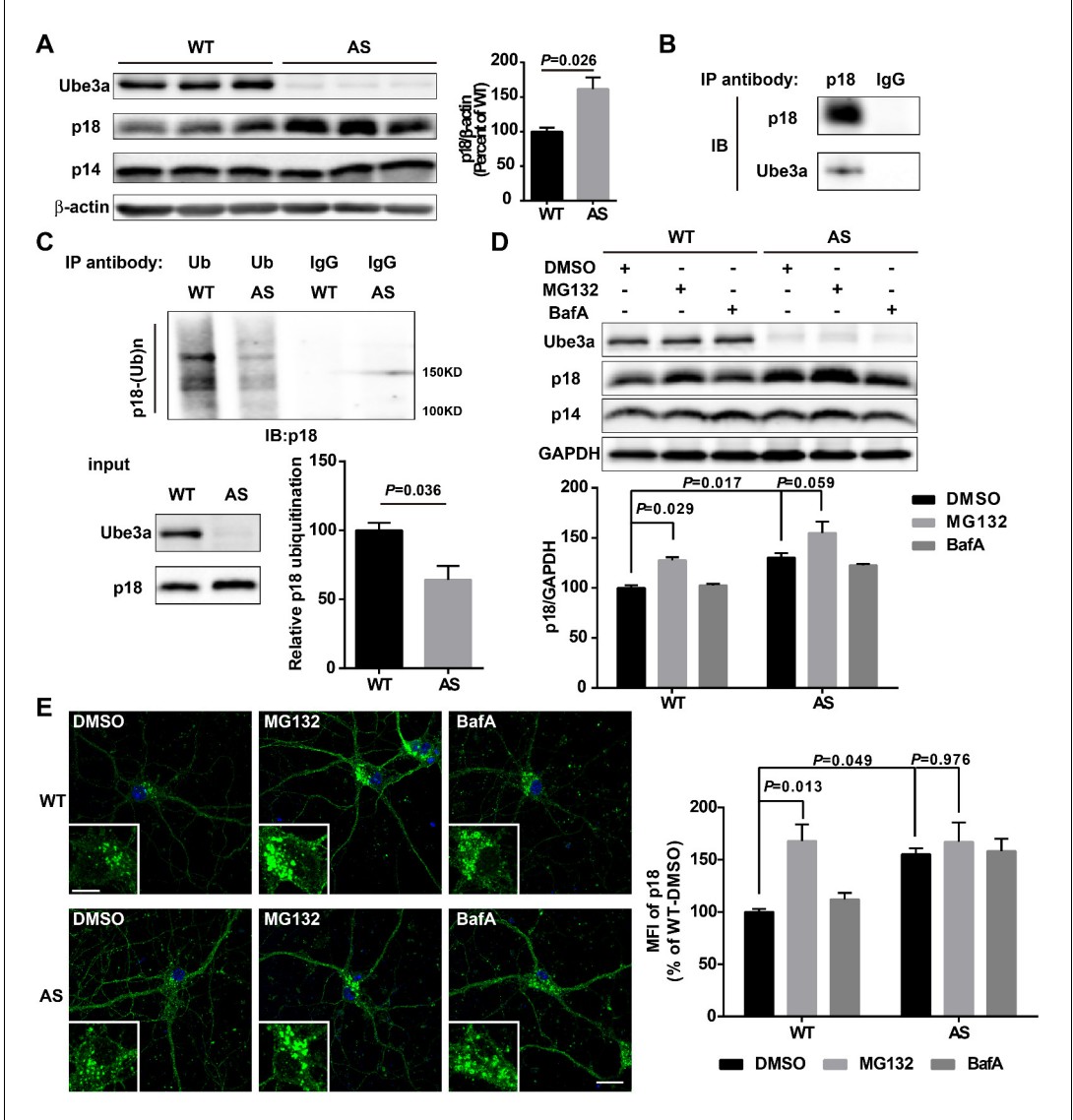

**Figure 3.** Ube3a regulates p18 levels in a proteasome-dependent manner in hippocampal neurons. (**A**) Left, Western blot analysis of p18 and p14 levels in crude membrane fractions (P2) of hippocampi from WT and AS mice. Right, quantitative analysis of blots. Results are expressed as % of values in WT mice and shown as means ± SEM N = 3 mice, p=0.026 (unpaired, two-tailed Student's t-test). (**B**) Interactions between Ube3a and p18 in hippocampal neuron cultures. Western blot analysis with anti-p18 and -Ube3a antibodies of immunoprecipitation performed with anti-p18 antibodies or control IgG. (**C**) Immunoprecipitation of hippocampal P2 fractions from WT and AS mice under denaturing conditions was performed with anti-ubiquitin antibodies or control IgG, and Western blots were labelled with anti-p18 antibodies. Ubiquitinated p18 proteins are indicated as 'p18-(Ub)n'. Lower left panel: levels of input proteins were evaluated by Western blots probed with Ube3a and p18 antibodies. Lower right panel: quantification of the relative abundance of ubiquitinated p18 in hippocampus of WT and AS mice (mean ± SEM, p=0.036 compared with WT mice, n = 3 mice, Student's t-test). (**D**) Effects of acute MG132 or bafilomycin A1 (BafA) treatment on p18 and p14 levels in hippocampus slices of WT and AS mice. Upper panel: representative Western blot images; lower panel: quantitative analysis of blots in upper panel. N = 3 independent experiments, p=0.029 WT/DMSO vs. WT/MG132, p=0.017 WT/DMSO vs. AS/DMSO, p=0.059 AS/DMSO vs. AS/MG132, two-way ANOVA with Tukey's post-test. (**E**) Representative images of p18 in WT and AS hippocampal neurons treated with DMSO, MG132, and BafA; insets: enlarged cell bodies. Right: Quantitative analysis of images. Data are expressed as mean ± SEM. N = 3 independent experiments, p=0.013 WT/DMSO vs. WT/MG132, p=0.049 WT/DMSO vs. AS/DMSO, p=0.976 AS/DMSO vs. AS/MG132; two-way ANOVA with Tukey's post hoc analysis. Scale bar = 20 and 10 μm in insets. See also *Figure 3—figure supplement 1* and *Figure 3—source data 1*.

DOI: https://doi.org/10.7554/eLife.37993.008

The following source data and figure supplement are available for figure 3:

**Source data 1.** Quantitative analyses of images and Western blots used for *Figure 3* and *Figure 3—figure supplement 1*.
DOI: https://doi.org/10.7554/eLife.37993.010

*Figure 3 continued on next page*

Figure 3 continued

**Figure supplement 1.** Levels of members of the Ragulator-Rag complex in WT and AS mice and in vivo denaturing immunoprecipitation assay of p18 ubiquitination.

DOI: https://doi.org/10.7554/eLife.37993.009

(*Figure 3C* and *Figure 3—figure supplement 1B*), indicating that the increase in p18 levels in AS mice likely results from a deficit in Ube3a-mediated p18 ubiquitination and degradation.

To determine whether Ube3a-mediated regulation of p18 was proteasome- and/or lysosome-dependent, acute hippocampal slices from WT and AS mice were treated with either a proteasome inhibitor, MG132 (10 μM), or a lysosome inhibitor, the vacuolar H$^+$-ATPase (V-ATPase) inhibitor, bafilomycin A1 (BafA, 100 nM), for 30 min. These concentrations and treatment duration have previously been shown to significantly inhibit proteasome or lysosomal function, respectively (*Kim et al., 2015*). As expected, levels of p18 were significantly higher in vehicle-treated slices from AS mice than in vehicle-treated slices from WT mice. Incubation of hippocampal slices with MG132, but not BafA, significantly increased p18 levels in WT slices and marginally in AS slices, possibly because of the residual expression of paternal Ube3a (*Figure 3D*). In addition, MG132 treatment (10 μM, 4 h) markedly increased p18 intensity in cultured hippocampal neurons from WT mice, while BafA treatment (100 nM, 4 h) only slightly increased p18 in cultured hippocampal neurons from both WT and AS mice (*Figure 3E*). These results strongly suggest that Ube3a decreases p18 levels via ubiquitination followed by proteasomal degradation.

## Increased p18 levels in AS mice are associated with increased lysosomal localization of the Ragulator-Rag complex and mTOR

Immunofluorescent staining showed that p18 was clearly co-localized with LAMP2 in CA1 pyramidal neurons (NeuN/LAMP2 double stain shown in *Figure 4—figure supplement 2A*), especially in cell bodies, in both WT and AS mice, and that more p18/LAMP2 double-stained puncta were detected in AS than in WT mice (*Figure 4A–B*). Similarly, lysosomal localization of other members of the Ragulator, LAMTOR4 (*Figure 4B* and *Figure 4—figure supplement 1A*), p14, and MP1 (*Figure 4—figure supplement 1D*) was also increased in CA1 pyramidal cell soma of AS compared with WT mice. Furthermore, dual immunohistochemical staining for either RagA/B or mTOR with LAMP2 showed that co-localization of these proteins with LAMP2 was markedly increased in AS mice compared with WT mice (*Figure 4B* and *Figure 4—figure supplement 1B–D*). p18 was also clearly co-localized with LAMP2 in apical dendrites in the hippocampal CA1 region of adult mice, and more p18/LAMP2 double-stained puncta were detected in AS than in WT mice (*Figure 4D* and *Figure 4—figure supplement 1E*). Similarly, more mTOR proteins were co-localized with LAMP2 in CA1 apical dendrites of AS than WT mice (*Figure 4C–D*). We next evaluated the co-localization of p-mTOR (Ser2448) with LAMP2. More p-mTOR/LAMP2 double-stained puncta were observed in both soma and apical dendrites in the hippocampal CA1 region of AS than in WT mice (*Figure 4E–F* and *Figure 4—figure supplement 1F*).

To further confirm the lysosomal localization of mTORC1, we examined the co-localization of Raptor with LAMP2. Raptor is a critical component of mTORC1 and serves as a scaffold to spatially position substrates in close proximity to mTOR (*Hara et al., 2002*; *Kim et al., 2002*; *Nojima et al., 2003*), and its binding to Rag GTPases is necessary and sufficient to activate mTORC1 (*Sancak et al., 2008*). Raptor was clearly co-localized with LAMP2 in CA1 pyramidal cell soma of adult mice, and more Raptor/LAMP2 double-stained puncta were detected in AS than in WT mice (*Figure 4—figure supplement 2B and D*). In contrast, Rictor, an essential component of mTORC2, showed no co-localization with LAMP2 in both WT and AS mice (*Figure 4—figure supplement 2C–D*), suggesting that mTORC2 may not be recruited to lysosomes. Consistently, Western blot results showed that levels of the Ragulator-Rag complex as well as those of mTOR and Raptor were markedly increased in lysosomal fractions of hippocampus from AS mice compared with WT mice (*Figure 4—figure supplement 2E*).

In addition to providing a platform for recruiting Rag GTPases and subsequently mTORC1 to lysosomes, p18 has also been shown to function as a RagA/B GEF, which facilitates the exchange of GDP from RagA/B to GTP (*Bar-Peled et al., 2012*). To test whether increased p18 levels in AS mice could lead to increased levels of GTP-bound RagA/B, we used the well-known co-

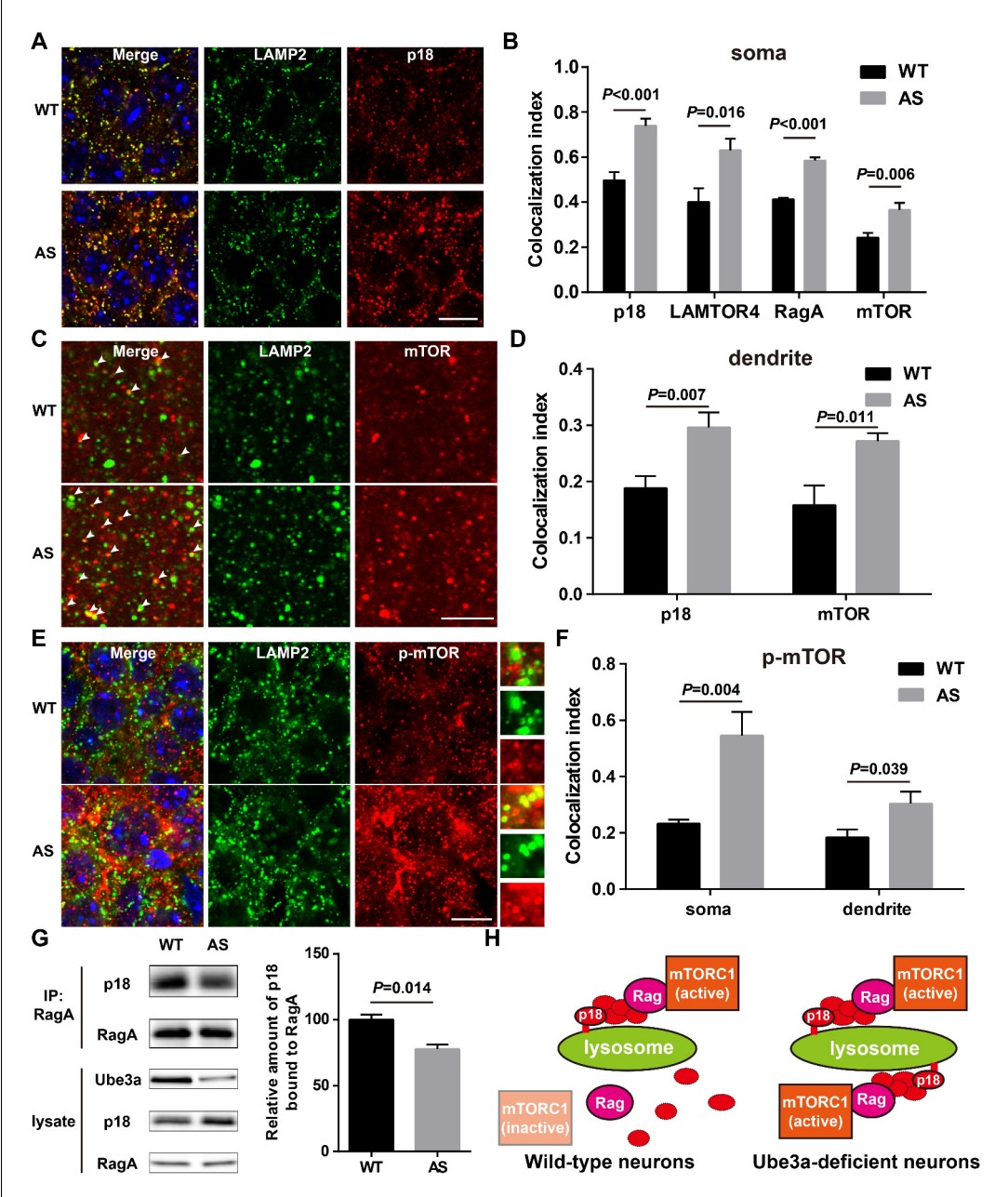

**Figure 4.** Lysosomal localization of the Ragulator-Rag complex and mTOR/p-mTOR in WT and AS mice. (A) Co-localization of p18 (red) with LAMP2 (green) in cell bodies of CA1 pyramidal neurons from WT and AS mice. Scale bar = 10 µm. (B) Quantification of p18-LAMP2 (n = 8 mice, p<0.001), LAMTOR4-LAMP2 (n = 6 mice, p=0.016), RagA-LAMP2 (n = 6 mice, p<0.001), and mTOR-LAMP2 (n = 8 mice, p=0.006) colocalization in cell bodies of CA1 pyramidal neurons from WT and AS mice shown in A and *Figure 4—figure supplement 1A—C*. Unpaired t-test. (C) Representative images of apical dendrites of CA1 pyramidal neurons stained with anti-mTOR (red) and -LAMP2 (green) antibodies. Arrowheads indicate puncta with dual staining. Scale bar = 5 µm. (D) Quantification of p18-LAMP2 (n = 8 mice, p=0.007) and mTOR-LAMP2 (n = 7 mice, p=0.011) co-localization in apical dendrites of CA1 pyramidal neurons from WT and AS mice. Unpaired t-test. (E) Co-localization of p-mTOR (red) with LAMP2 (green) in cell bodies of CA1 pyramidal neurons from WT and AS mice. Scale bar = 10 µm. Insets show selected fields that were magnified 10 times. (F) Quantification of p-mTOR-LAMP2 co-localization in cell bodies (p=0.004) and dendrites (p=0.039) of CA1 pyramidal neurons from WT and AS mice. N = 6 mice, unpaired t-test. (G) Homogenates from WT and AS mouse hippocampus were immunoprecipitated with an anti-RagA antibody and probed with the indicated antibodies. Right, quantification of the relative abundance of p18 bound to RagA (mean ± SEM, p=0.014, n = 3 mice, Student's t-test). (H) Model proposing that the Ragulator interacts with Rag, which in turn recruits mTORC1 to be activated on lysosomes in neurons. In Ube3a-deficient neurons, increased Ragulator-Rag complex on lysosomes results in mTORC1 over-activation. See also *Figure 4—figure supplements 1* and *2* and *Figure 4—source data 1*.

*Figure 4 continued on next page*

*Figure 4 continued*

DOI: https://doi.org/10.7554/eLife.37993.011

The following source data and figure supplements are available for figure 4:

**Source data 1.** Quantitative analyses of images and Western blots used for *Figure 4* and *Figure 4—figure supplements 1* and *2*.
DOI: https://doi.org/10.7554/eLife.37993.014
**Figure supplement 1.** Lysosomal localization of members of the Ragulator-Rag complex and mTOR/p-mTOR in the hippocampus of WT and AS mice.
DOI: https://doi.org/10.7554/eLife.37993.012
**Figure supplement 2.** Lysosomal localization of Raptor and Rictor in the hippocampus of WT and AS mice, and Western blot analysis of Ragulator-Rag complex as well as mTOR and Raptor in WT and AS hippocampal lysosome fractions.
DOI: https://doi.org/10.7554/eLife.37993.013

immunoprecipitation assay of p18 and RagA, based on the observation that GTP-bound RagA/B has a lower affinity for the Ragulator compared with GDP-bound RagA/B (*Bar-Peled et al., 2012*; *Castellano et al., 2017*). Co-immunoprecipitation results showed that levels of p18 immunoprecipitated by RagA antibodies were significantly lower in samples from AS mice compared with WT mice (*Figure 4G*). Collectively, these results showed that increased p18 levels in the hippocampus of AS mice facilitate lysosomal anchoring of the Ragulator-Rag complex and activation of mTORC1 (see schematic in *Figure 4H*).

## p18 KD counteracts Ube3a deficiency-induced abnormal mTOR signaling and changes in dendritic spine morphology and actin polymerization in cultured hippocampal neurons

We next directly tested whether reducing p18 levels could modify Ube3a deficiency-induced mTORC1 over-activation in cultured hippocampal neurons. p18 expression was reduced by infection with a set of p18 shRNA lentiviruses, while Ube3a KD was achieved with Accell Ube3a siRNA (*Figure 5A–B*). Ube3a KD resulted in increased p18 levels (*Figure 5A–B*), in parallel with increased mTORC1 activation, as reflected by increased phosphorylation of mTOR and its downstream substrate S6 and 4EBP1, and decreased mTORC2 activation, as reflected by decreased (p-)PKCα levels and phosphorylation of AKT. Increased mTORC1 activation and decreased mTORC2 activation were reversed by p18 KD (*Figure 5A–B* and *Figure 5—figure supplement 1A*). These results strongly suggest that increased p18 levels contribute to Ube3a deficiency-induced abnormal mTOR signaling in AS mice. Importantly, p18 shRNA KD in the absence of Ube3a KD led to a reduction in mTORC1 activation and a concomitant increase in the activity of mTORC2 (*Figure 5A,B*), which further underscores the notion that mTOR signaling is very sensitive to changes in p18 levels.

We also determined whether Ube3a-mediated p18 regulation could affect dendritic spine morphology and actin polymerization. Cultured hippocampal neurons from AS or WT mice were co-infected with p18 shRNA or control shRNA lentiviruses and a GFP control lentivirus, and actin polymerization was determined by staining for filamentous actin (F-actin). Confocal images of infected neurons indicated that p18 shRNA infection reduced p18 expression in cultured neurons from both WT and AS mice (*Figure 5—figure supplement 1B*). Neurons from AS mice exhibited reduced dendritic spine density and actin polymerization compared with neurons from WT mice (*Figure 5C,D*). p18 shRNA KD increased spine density and restored actin polymerization in neurons from AS mice, and slightly enhanced spine density and F-actin levels in neurons from WT mice (*Figure 5C,D*). These results indicate that deficiency in Ube3a-mediated p18 degradation contributes to spine defects and actin polymerization abnormality in neurons from AS mice. The effects of p18 KD on spine maturation were further analyzed in in vivo studies presented in the following sections.

## p18 KD promotes LTP and stimulates dendritic spine maturation in AS mice

High magnification confocal images of adult hippocampal CA1 pyramidal neurons revealed that in addition to being co-localized with lysosomal markers, p18 was also localized in the vicinity of and often co-localized with PSD95 (arrowheads in *Figure 6A*). Quantitative analysis showed that the number of p18-immunoreactive puncta was markedly increased in AS mice (*Figure 6—figure supplement 1A*). Furthermore, the percentage of PSD95-stained puncta labeled with p18 antibodies

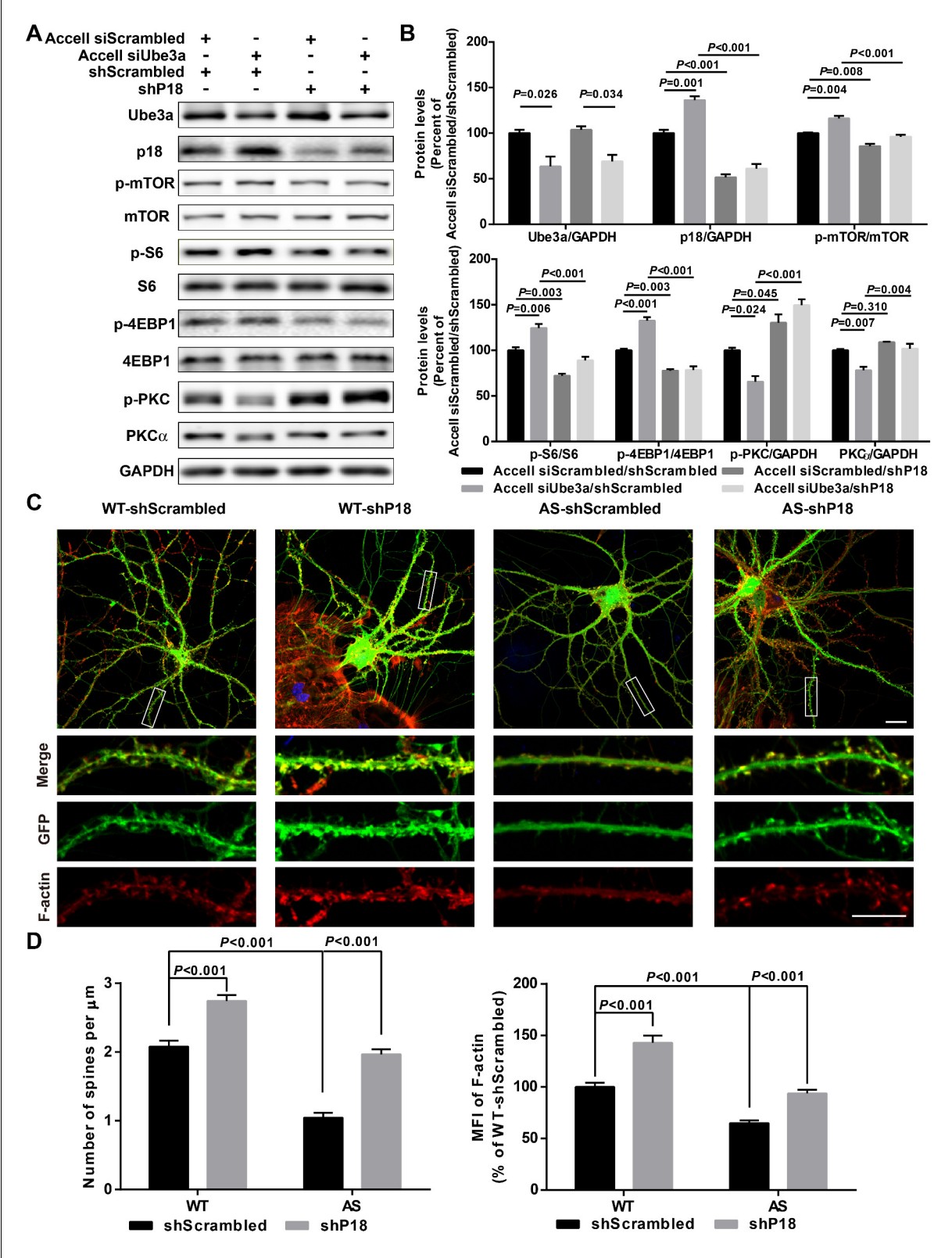

**Figure 5.** p18 mediates the effects of Ube3a on mTOR signaling, dendritic spine morphology, and actin polymerization. (**A**) Representative images of Western blots labeled with Ube3a, p18, p-mTOR, mTOR, p-S6, S6, p-4EBP1, 4EBP1, p-PKC, and PKCα (GAPDH as a loading control). Protein lysates from cultured hippocampal neurons transfected with the indicated constructs were prepared for Western blot analysis. (**B**) Quantitative analysis of blots shown in (**A**). N = 3 independent experiments, Accell siScrambled/shScrambled vs. Accell siUbe3a/shScrambled, p=0.026 (Ube3a), p=0.001 (p18),

*Figure 5 continued on next page*

*Figure 5 continued*

p=0.004 (p-mTOR), p=0.006 (p–S6), p<0.001 (p-4EBP1), p=0.024 (p-PKC), p=0.007 (PKCα); Accell siScrambled/shScrambled vs. Accell siScrambled/shP18, p<0.001 (p18), p=0.008 (p-mTOR), p=0.003 (p–S6), p=0.003 (p-4EBP1), p=0.045 (p-PKC), p=0.310 (PKCα); Accell siUbe3a/shScrambled vs. Accell siUbe3a/shP18, p<0.001 (p18), p<0.001 (p-mTOR), p<0.001 (p–S6), p<0.001 (p-4EBP1), p<0.001 (p-PKC), p=0.004 (PKCα); Accell siScrambled/shP18 vs. Accell siUbe3a/shP18, p=0.034 (Ube3a); two-way ANOVA with Tukey's post-test. (C) Representative images of F-actin (red) and GFP in cultured WT and AS hippocampal neurons (22 DIV) co-infected with GFP lentivirus and p18 shRNA or scrambled shRNA lentivirus. Scale bar, 20 µm (upper) or 10 µm (lower). (D) Quantitative analysis of images shown in (C). N = 9 neurons from at least three independent experiments, p<0.001, two-way ANOVA with Tukey's post-test. See also *Figure 5—figure supplement 1* and *Figure 5—source data 1*.

DOI: https://doi.org/10.7554/eLife.37993.015

The following source data and figure supplement are available for figure 5:

**Source data 1.** Quantitative analyses of images and Western blots used for *Figure 5* and *Figure 5—figure supplement 1*.
DOI: https://doi.org/10.7554/eLife.37993.017
**Figure supplement 1.** Effects of p18 knockdown in WT and AS hippocampal neurons.
DOI: https://doi.org/10.7554/eLife.37993.016

was also significantly increased in AS mice compared with WT mice (*Figure 6—figure supplement 1A*), although there was no significant difference in the overall number of PSD95-immunopositive puncta between AS and WT mice (*Figure 6—figure supplement 1A*). To determine whether increased synaptic p18 levels could contribute to impaired functional and structural synaptic plasticity in AS mice, we performed in vivo p18 siRNA KD experiments.

AAV vectors containing p18 siRNA or scrambled siRNA (control) were bilaterally injected into the dorsal hippocampal CA1 region of WT and AS mice (*Figure 6—figure supplement 1B*), and LTP in hippocampal slices was analyzed 4 weeks later from these four experimental groups. To evaluate infection efficiency, GFP expression, as well as p18 levels were determined in hippocampal slices by immunohistochemistry following LTP recording (*Figure 6B*). Only mice that exhibited significant GFP expression in the CA1 region (*Figure 6—figure supplement 1C*) were included in the LTP and spine analyses. Quantitative analysis showed that p18 expression was significantly increased in AS mice compared with WT mice, and this increase was reversed by p18 siRNA infection (*Figure 6C*). p18 siRNA infection also significantly reduced p18 expression in WT mice, to a larger degree than in AS mice (*Figure 6C*). Levels of p18, p-mTOR/mTOR, p-S6/S6, and PKCα in CA1 regions were also determined using Western blots; p18 KD reduced mTORC1 activity and increased mTORC2 activity in both WT and AS mice (*Figure 6—figure supplement 1D,E*).

Baseline synaptic responses, including input/output curves (I/O curves) and paired-pulse facilitation, were not altered by control siRNA or p18 siRNA in either AS or WT mice (*Figure 6—figure supplement 2*). LTP was induced by applying theta-burst stimulation (TBS) to Schaffer collaterals in the CA1 region, as previously described (*Baudry et al., 2012*; *Sun et al., 2015b*). TBS elicited typical LTP in field CA1 of hippocampal slices from control siRNA-injected WT mice, whereas it only elicited a transient facilitation in slices from control siRNA-injected AS mice (*Figure 6D,E*), a result similar to that found in slices from untreated AS mice (*Baudry et al., 2012*; *Sun et al., 2015b*). In contrast, bilateral CA1 injection of p18 siRNA enhanced TBS-elicited LTP in hippocampal slices from AS mice (*Figure 6D,E*), while it reduced TBS-induced LTP in slices from WT mice (*Figure 6D,E*).

To determine whether reduced LTP in the p18 siRNA WT group resulted from reduced mTORC1 activation because of 'sub normal' p18 levels, we used an mTOR activator MHY1485. Pre-incubation of hippocampal slices with MHY1485 (2 µM) for 60 min reestablished TBS-elicited LTP to levels identical to those in control siRNA-injected WT mice (*Figure 7A,B*). Levels of p18, p-mTOR/mTOR, and p-S6K1/S6K1 in AAV infected regions were determined using Western blots; p18 KD-induced reduction of mTORC1 signaling in WT mice was reversed by acute treatment with MHY1485 (*Figure 7C*). We also analyzed Arc (activity-regulated cytoskeleton-associated protein) levels in CA1 dendritic field injected with control or p18 siRNA, as Arc is a well-known downstream effector of Ube3a and/or mTOR signaling (*Greer et al., 2010*; *Sun et al., 2016*). As expected, levels of Arc were significantly higher in control siRNA-injected AS mice compared with control siRNA-injected WT mice; p18 KD significantly reduced Arc expression in both WT and AS hippocampal slices (*Figure 7D,E*). Of note, p18 KD induced a larger reduction in Arc expression in WT mice than in AS mice (*Figure 7D, E*), which may contribute to reduced LTP in the p18 siRNA WT group. Together, these results suggest that p18-mediated regulation of Arc levels is critical for LTP in WT and AS mice.

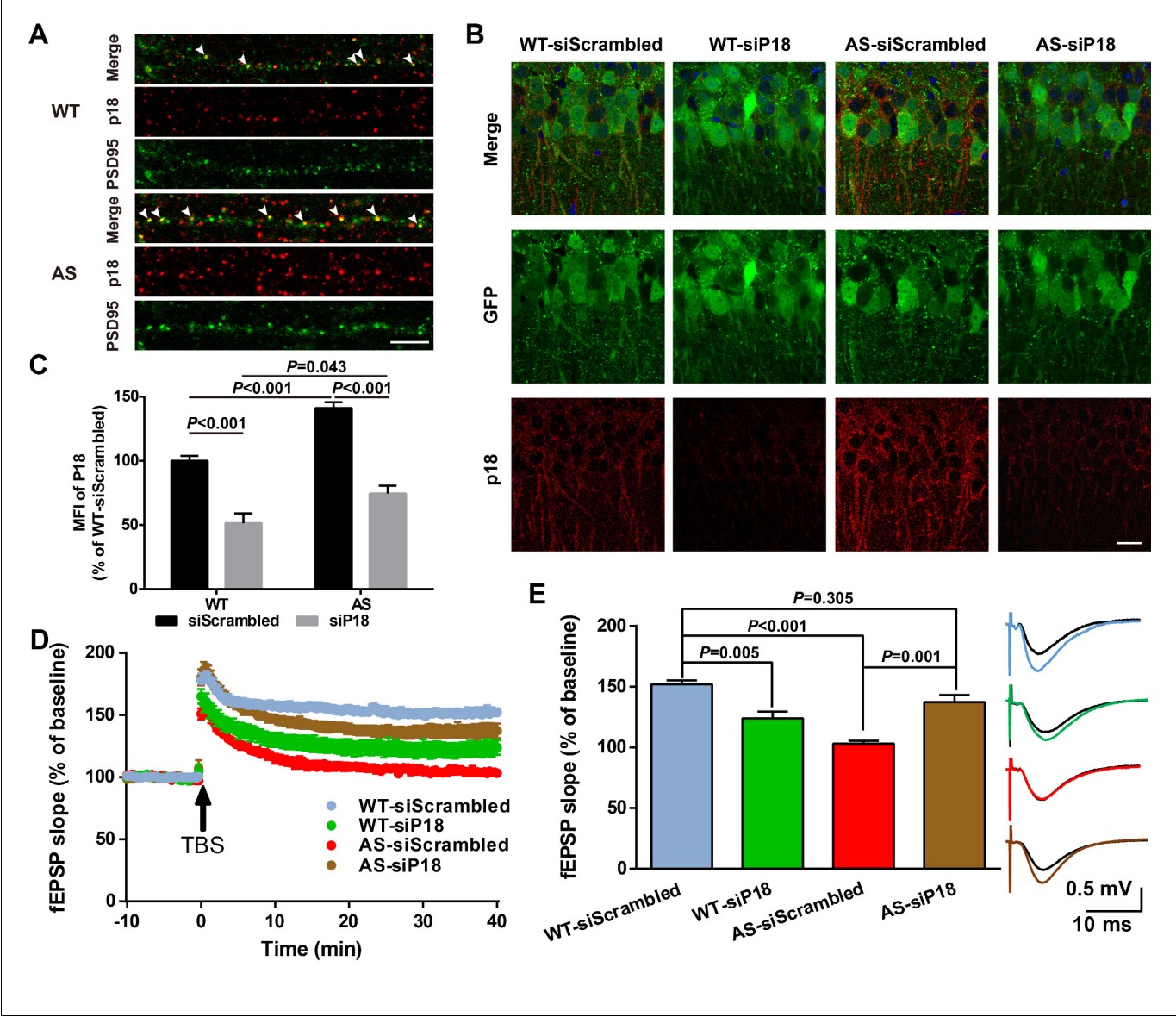

**Figure 6.** Effects of p18 KD in hippocampal CA1 region on LTP in WT and AS mice. (**A**) Representative images of dendrites of CA1 pyramidal neurons stained with anti-p18 (red) and -PSD95 (green) antibodies. Arrowheads indicate co-localized puncta. Scale bar = 10 μm. (**B**) Representative images of CA1 pyramidal neurons stained with anti-p18 (red) and -GFP (green) antibodies. Scale bar = 20 μm. (**C**) Quantitative analysis of the mean fluorescence intensity (MFI) of p18-immunoreactive puncta in GFP-positive CA1 pyramidal neurons. N = 6 mice, p<0.001, WT-siScrambled vs. WT-siP18; p<0.001, WT-siScrambled vs. AS-siScrambled; p<0.001, AS-siScrambled vs. AS-siP18; p=0.043, WT-siP18 vs. AS-siP18, two-way ANOVA with Tukey's post-test. (**D, E**) Effects of AAV siRNA-mediated p18 KD on LTP in WT and AS mice. (**D**) Slopes of fEPSPs were normalized to the average values recorded during the 10 min baseline. (**E**) Means ± SEM of fEPSPs measured 40 min after TBS in different groups. N = 7–14 slices from four to eight mice, p=0.005, WT-siScrambled vs. WT-siP18, p<0.001, WT-siScrambled vs. AS-siScrambled, p=0.001, AS-siScrambled vs. AS-siP18, p=0.305, WT- siScrambled vs. AS-siP18, two-way ANOVA with Tukey's post-test. Inset shows representative traces of evoked fEPSPs before and 40 min after TBS. Scale bar 0.5 mV/10 ms. See also *Figure 6—figure supplements 1* and *2* and *Figure 6—source data 1*.

DOI: https://doi.org/10.7554/eLife.37993.018

The following source data and figure supplements are available for figure 6:

**Source data 1.** Source data for *Figure 6* and *Figure 6—figure supplement 1*.
DOI: https://doi.org/10.7554/eLife.37993.021
**Figure supplement 1.** Effects of p18 knockdown in hippocampal CA1 region on mTOR signaling in WT and AS mice.
DOI: https://doi.org/10.7554/eLife.37993.019
**Figure supplement 2.** Effects of p18 knockdown in hippocampal CA1 region on input/output curves and paired-pulse facilitation in WT and AS mice.
DOI: https://doi.org/10.7554/eLife.37993.020

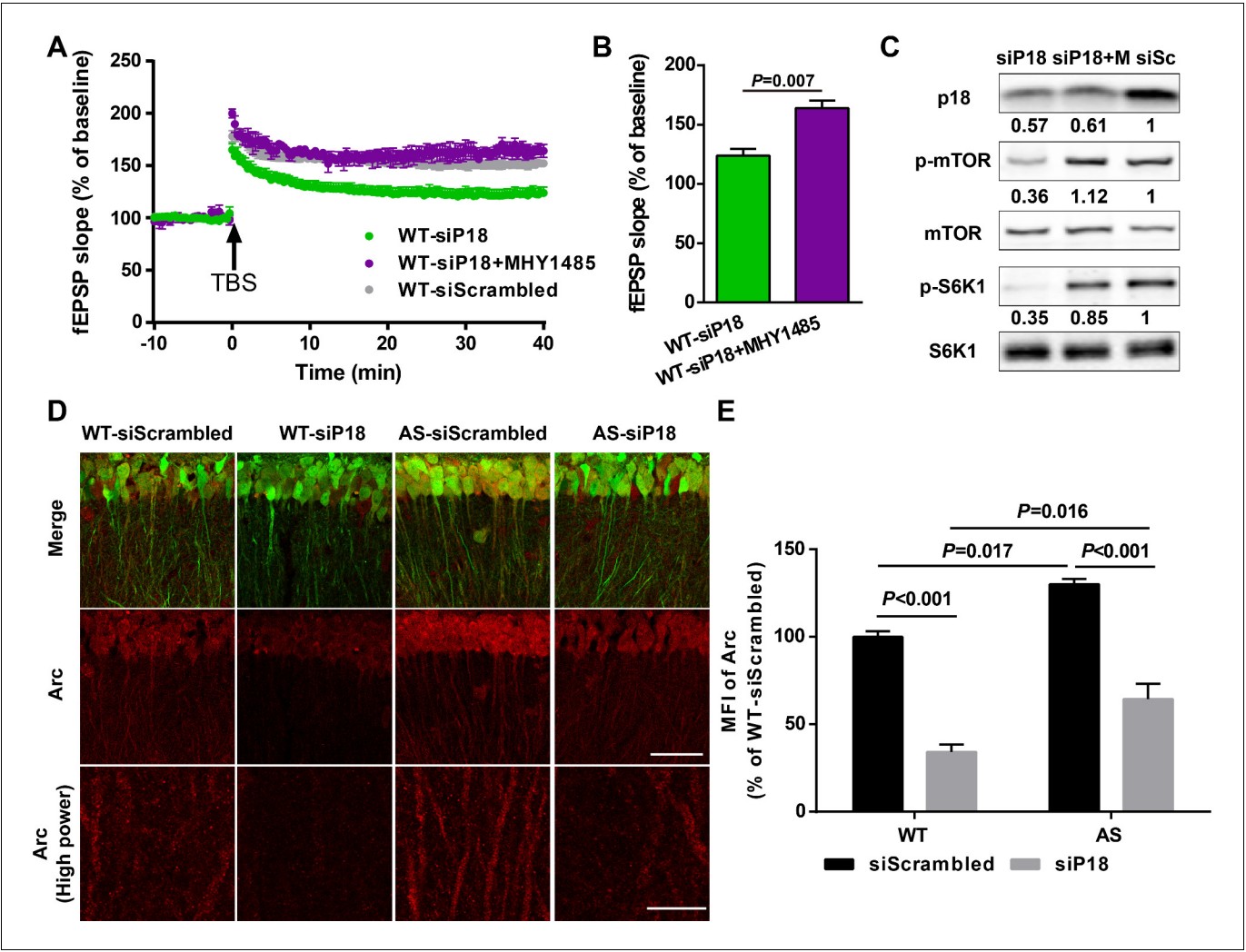

**Figure 7.** p18 KD impairs LTP in WT neurons as a result of over-inhibition of mTORC1 activity and Arc levels. (A–C) Effects of MHY1485 treatment on LTP in p18 siRNA-injected WT mice. (A) Slopes of fEPSPs were normalized to the average values recorded during the 10 min baseline. (B) Means ± SEMof fEPSPs measured 40 min after TBS in different groups. N = 3–14 slices from three to eight mice, p=0.007, unpaired t-test. (C) Representative Western blots showing the relative abundance of p18, p-mTOR/mTOR, and p-S6K/S6K in lysates from control siRNA (siSc) or p18 siRNA (siP18)-infected WT hippocampal slices. Slices were treated with or without MHY1485 (M). (D,E) Effects of Ube3a deficiency and p18 KD in the hippocampal CA1 region on Arc expression. (D) Representative images of CA1 pyramidal neurons stained with anti-Arc (red) and -GFP (green) antibodies. Scale bar = 50 μm (low power images) and 10 μm (high power images). (E) Quantitative analysis of the MFI of Arc-immunoreactivty of CA1 pyramidal neurons (means ± SEM of 3 slices from three different animals; p<0.001, WT-siScrambled vs. WT-siP18; p=0.017, WT-siScrambled vs. AS-siScrambled; p<0.001, AS-siScrambled vs. AS-siP18; p=0.016, WT-siP18 vs. AS-siP18, two-way ANOVA with Tukey's post-hoc analysis). See also *Figure 7—source data 1*.

DOI: https://doi.org/10.7554/eLife.37993.022

The following source data is available for figure 7:

**Source data 1.** Source data for *Figure 7*.

DOI: https://doi.org/10.7554/eLife.37993.023

We then performed Golgi staining in the hippocampal CA1 region of WT and AS mice injected with p18 siRNA or control siRNA. As previously reported (*Dindot et al., 2008*; *Sun et al., 2016*), spine density was lower with a higher proportion of immature spines (thin or filopodia) in AS mice compared with WT mice (*Figure 8A,B* and *Figure 8—figure supplement 1A,B*). p18 KD significantly increased the number of mature dendritic spines in hippocampal pyramidal neurons of AS mice, while it increased the number and proportion of immature dendritic spines of WT mice (*Figure 8A,B* and *Figure 8—figure supplement 1A,B*). The effects of p18 KD on the number of

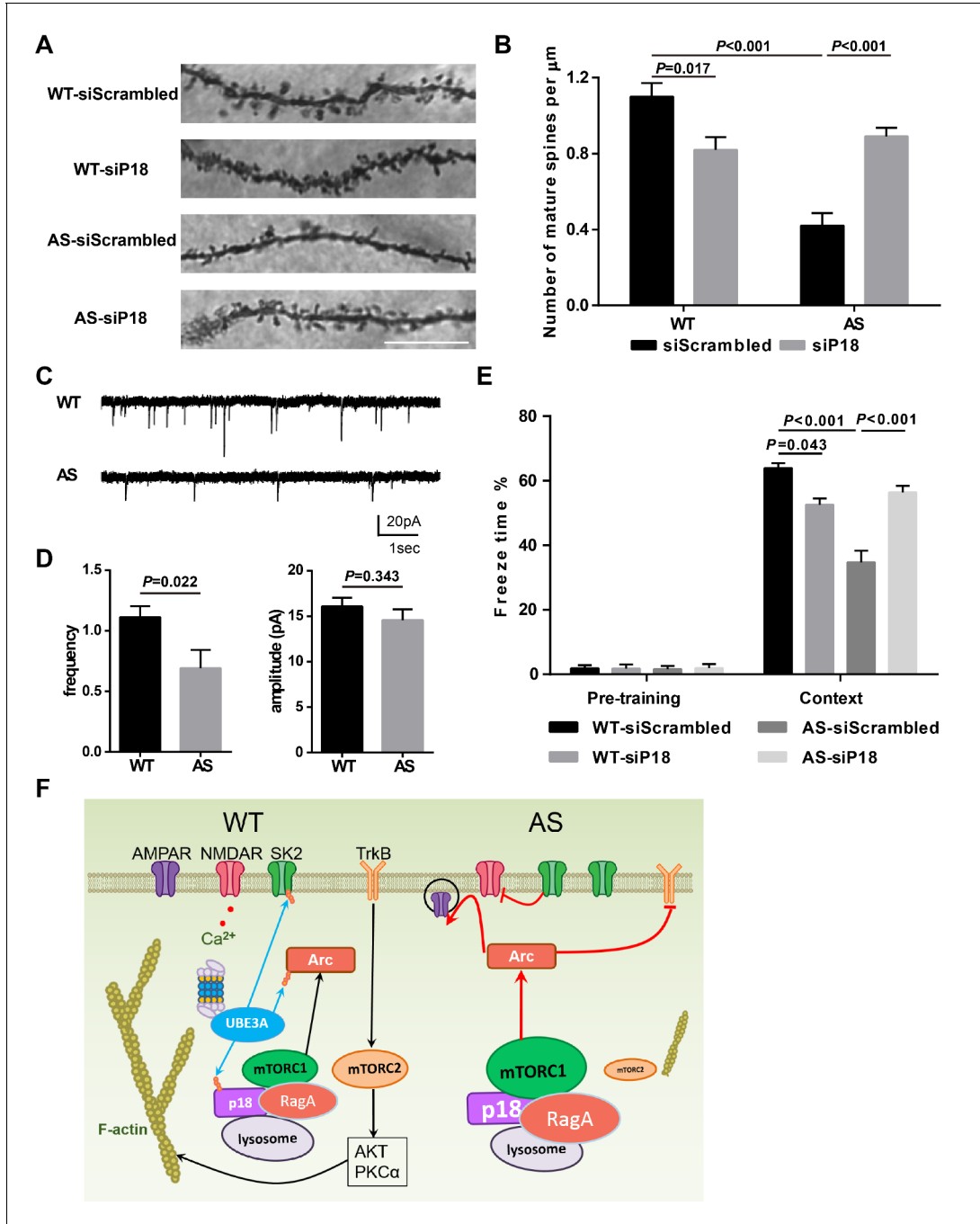

**Figure 8.** Effects of Ube3a deficiency and p18 KD in hippocampal CA1 region on dendritic spine morphology, mEPSCs, and fear-conditioning memory. (A) Representative light micrograph images from Golgi-impregnated CA1 pyramidal neurons. Scale bar = 10 µm. (B) Quantitative analysis of mature dendritic spine (multi-head, mushroom, and stubby spines) density shown in (A) (means ±SEM from 10 slices). p=0.017, WT-siScrambled vs. WT-siP18; p<0.001, WT-siScrambled vs. AS-siScrambled; p<0.001, AS-siScrambled vs. AS-siP18, two-way ANOVA with Tukey's post-test. (C) Representative mEPSC traces recorded in hippocampal neurons from WT and AS slices. Scale bar, 20 pA/1 s. (D) Quantification of mEPSC frequency (p=0.022) and amplitude (p=0.343) from WT (n = 12) and AS (n = 7) mice. Student's t-test. (E) % freezing for different experimental groups in context memory (means ± SEM of 6–10 mice; p=0.043, WT-siScrambled vs. WT-siP18; p<0.001, WT-siScrambled vs. AS-siScrambled; p<0.001, AS-siScrambled vs. AS-siP18, two-way ANOVA with Tukey's post-hoc analysis). (F) Model for Ube3a-mediated regulation of synaptic plasticity (see text for details). See also *Figure 8—figure supplement 1* and *Figure 8—source data 1*.

DOI: https://doi.org/10.7554/eLife.37993.024

The following source data and figure supplement are available for figure 8:

**Source data 1.** Source data for *Figure 8* and *Figure 8—figure supplement 1*.

*Figure 8 continued on next page*

*Figure 8 continued*

DOI: https://doi.org/10.7554/eLife.37993.026

**Figure supplement 1.** Effects of p18 knockdown in hippocampal CA1 region on dendritic spine number and the proportion of various spine types as well as tone memory in WT and AS mice.

DOI: https://doi.org/10.7554/eLife.37993.025

mature dendritic spines (*Figure 8B*) correlated well with the effects on LTP. As an independent means of assessing the effect of changes in spine morphology, we recorded miniature excitatory postsynaptic currents (mEPSCs) from WT and AS hippocampal pyramidal neurons in acute slice preparations. We observed a reduction in the frequency and no change in the amplitude of mEPSCs in AS neurons compared with WT neurons (*Figure 8C,D*).

## p18 KD improves learning and memory performance of AS mice in the fear-conditioning paradigm

To determine whether p18 KD could also ameliorate the impairment in hippocampus-dependent learning in AS mice, we analyzed fear conditioning in AS and WT mice 8 weeks after AAV injection. AS mice were impaired in context- and tone-dependent fear-conditioning, and p18 KD in the CA1 region significantly enhanced context-dependent learning performance, while it did not affect tone-dependent learning in AS mice (*Figure 8E* and *Figure 8—figure supplement 1C*). Consistent with its effects on LTP and spine morphology, p18 siRNA treatment reduced context-dependent learning performance in WT mice (*Figure 8E*). There was no difference in freezing time during the pre-conditioning period, or before tone application in the testing period between all experimental groups (*Figure 8E* and *Figure 8—figure supplement 1C*).

## Discussion

The last decade has witnessed a rapid growth in our knowledge of amino acid-mediated regulation of mTORC1 signaling, including the identification of a lysosome-based platform for its activation. Although it has been shown that p18 has a critical role in stabilization of the Ragulator-Rag complex (*de Araujo et al., 2017*) and anchoring it to lysosomes, little is known regarding the regulation of p18 levels. Our results provide several lines of evidence indicating that Ube3a is an E3 ligase for p18, and that Ube3a-mediated p18 ubiquitination leads to its degradation by the proteasome. Of note, p18 myristoylation and lysosomal localization of p18 were required for its ubiquitination and mTORC1 activation, suggesting that Ube3a specifically targets lysosomal-localized p18, thereby efficiently regulating lysosome-based mTORC1 activation.

Although regulation of mTORC1 by the TSC complex-Rheb axis is well documented in the CNS, compared with various cell lines, its regulation by a lysosome-anchored platform has rarely been studied. Our results showed for the first time that, in hippocampal neurons, p18 is essential for lysosomal localization of other Ragulator members and Rag GTPases, as p18 KD markedly reduced the lysosomal localization of these proteins, which is in agreement with that reported in other cell types (*Nada et al., 2009*; *Sancak et al., 2010*). We further showed that lysosomal localization of the Ragulator-Rag complex is essential for mTORC1 activation in hippocampal neurons. Additionally, we showed that lysosomal localization and mTORC1 activation depends on Ube3a-mediated regulation of p18 levels, as a Ube3a deficiency-induced increase in p18 levels enhanced lysosomal recruitment of the Ragulator-Rag complex, leading to mTORC1 over-activation. Our results also support the idea that the Ragulator functions not only as a platform but also as a Rag GTPase GEF to facilitate mTORC1 activation (*Bar-Peled et al., 2012*). Furthermore, p18 KD reversed Ube3a deficiency-induced increase in mTORC1 activation and decrease in mTORC2 activation. Collectively, our results indicate that Ube3a-mediated p18 ubiquitination and degradation are critical for maintaining an optimal level of lysosome-anchored Ragulator-Rag complex and mTORC1 activation in hippocampal neurons.

What could be the functional roles of lysosome-based/p18-dependent regulation of mTOR signaling in hippocampal neurons? In contrast to the classical notion that lysosomes are mostly localized in neuronal soma and function merely as a 'recycling device', recent studies have shown that lysosomes are enriched in dendrites and that activity-dependent release of lysosomal hydrolases plays

important roles in synaptic plasticity (*Goo et al., 2017*). We showed here that p18 was not only co-localized with lysosome markers, but was also present in the vicinity of or co-localized with PSD95-labeled synapses in apical dendrites of hippocampal CA1 pyramidal neurons. This localization enables p18 to rapidly assemble or disassemble lysosome-based platforms for mTORC1 activation, and to possibly regulate other enzymes (e.g. MARK), in the vicinity of synapses, thereby regulating synaptic plasticity. Our results also underscored the importance of maintaining optimal p18 levels, as both 'too much' and 'too little' p18 resulted in abnormal spine structure and synaptic plasticity. Specifically, we showed that Ube3a deficiency-induced increase in p18 levels was associated with increased mTORC1 activation, decreased spine maturation, and impaired LTP and learning performance, all of which were reversed by p18 KD. On the other hand, p18 KD in WT mice resulted in an increased number of immature spines, LTP impairment, and reduced learning performance, possibly because of decreased mTORC1 signaling, as an mTORC1 activator rescued p18 KD-induced LTP impairment. Different levels of mTORC1 inhibition could also explain why in our previous study rapamycin treatment improved LTP and spine maturation in AS mice, but did not affect either LTP or spine morphology in WT mice (*Sun et al., 2016*). Along this line, work from the Costa-Mattioli lab clearly indicated that a low concentration of rapamycin has no effect on LTP, while a high concentration of rapamycin impairs LTP in WT mice (*Stoica et al., 2011*). Similarly, it has been reported that although ErbB inhibitors enhanced contextual fear memory in AS mice, they impaired long-term memory in WT mice (*Kaphzan et al., 2012*).

Several downstream mechanisms could account for the effects of different levels of p18-mTORC1 signaling in synaptic plasticity and learning and memory (see schematic in *Figure 8F*). First, activation of mTORC1-dependent local protein synthesis could account for the increased levels of the immediate-early gene product, Arc in AS mice reported by us and other groups (*Cao et al., 2013*; *Greer et al., 2010*; *Sun et al., 2016*). p18 KD reduced Arc immunoreactivity in CA1 pyramidal neurons, possibly through inhibition of the mTORC1-S6K1 pathway, as was the case with rapamycin treatment (*Sun et al., 2016*). Arc is known to induce AMPAR endocytosis, which dampens LTP expression and favors LTD induction (*Chowdhury et al., 2006*; *Rial Verde et al., 2006*). Related to this, both NMDAR-dependent and mGluR-dependent LTD in hippocampus are enhanced in AS mice (*Pignatelli et al., 2014*; *Sun et al., 2015b*). Increased Arc levels have been found to impair TrkB-PSD95 signaling in AS mice (*Cao et al., 2013*). On the other hand, inhibition of mTORC1 activity could also inhibit spine maturation, as mTORC1 activation has been shown to interact with other local synaptic events to promote spine enlargement (*Henry et al., 2017*).

Second, ample evidence has indicated that the dynamic properties of actin networks are crucial for synaptogenesis and synaptic plasticity; LTP consolidation is accompanied by increased levels of F-actin (*Kramár et al., 2006*; *Lin et al., 2005*). mTORC2 activity, which is reduced by mTORC1-mediated feedback inhibition in AS mice (*Sun et al., 2015a*), has been shown to be crucial for actin polymerization (*He et al., 2013*; *Huang et al., 2013*; *Jacinto et al., 2004*; *Sun et al., 2016*; *Thomanetz et al., 2013*). LTP impairment in AS mice was associated with reduced TBS-induced actin polymerization, compared with WT mice, and this reduction could be ameliorated by either a positive AMPAR modulator or a SK2 channel blocker (*Baudry et al., 2012*; *Sun et al., 2015b*). Both compounds increase NMDAR activity and $Ca^{2+}$ influx, thereby activating signaling proteins (e.g. CamKII, PKA, Rho), which facilitate F-actin formation. Similarly, reducing inhibitory inputs to CA1 pyramidal neurons using an ErbB inhibitor (*Kaphzan et al., 2012*), could also facilitate the activation of these signaling pathways and spine remodeling. p18 KD increased mTORC2 activity, resulting in actin polymerization and spine maturation, albeit through different downstream signaling pathways (activation of PKCα and Akt, etc.). Conditional deletion of *Rictor* in the postnatal murine forebrain greatly reduced mTORC2 activity and dendritic spine density in CA1 pyramidal neurons (*Huang et al., 2013*). Finally, p18 has been shown to directly interact with p27 (kip1), thereby regulating RhoA activity and actin remodeling (*Hoshino et al., 2011*), and autophagic activity (*Zada et al., 2015*). Whether these mTOR-independent p18 functions play any role in synaptic plasticity and brain development remains to be determined. Of note, baseline synaptic transmission and paired-pulse facilitation were not altered by p18 KD in both WT and AS mice, indicating that changes in synaptic plasticity resulting from p18 KD are likely a result of postsynaptic modifications related to processes that promote actin filament assembly during the minutes following TBS. Our results also indicated that, while there was a significant reduction in the frequency of mEPSCs in AS mice, mEPSC amplitude was not different from that found in WT mice, a result in agreement with

that of *Greer et al. (2010)*, but not that of *Kaphzan et al. (2012)*. This pattern would be consistent with a loss of mature spines and the existence of a relatively normal AMPA receptor density in the remaining intact spines of pyramidal neurons of AS mice.

Deregulation of mTOR signaling has been identified as a phenotypic feature common to various forms of ASD, including fragile X syndrome, and mutations in tuberous sclerosis complex 1 and 2 (*TSC1/2*), neurofibromatosis 1, and phosphatase and tensin homolog (*PTEN*) (*Huber et al., 2015*). In contrast to our findings, *Tang et al. (2014)* recently reported that over-activation of mTORC1 in $TSC2^{+/-}$ ASD mice resulted in increased spine density because of inhibition of the autophagy that underlies postnatal spine pruning. However, to date there is no report indicating that there is decreased autophagy in AS mice, suggesting the existence of different mechanisms downstream of mTORC1 in these two different mouse models. Although mTOR signaling is increased in Fragile X mouse models, a recent report showed that chronic rapamycin treatment did not reverse behavioral phenotypes and had adverse effects on sleep and social behavior in both control and *Fmr1* KO mice (*Saré et al., 2017*). These results strengthen the notion that further understanding of the mTOR pathway and its upstream and downstream regulation is needed.

Although we propose that Ube3a-mediated regulation of the p18-mTOR pathway is crucial in the pathogenesis of AS, our work by no means intends to conclude that p18 is the sole Ube3a target implicated in AS. Rather, our results indicate that the newly identified regulation of mTORC1 activation by lysosome-located p18 is present in the brain and plays important roles in synaptic plasticity, and document the existence of another layer of regulation in the already complex mTOR pathway, namely the regulation of p18 levels by Ube3a. Importantly, while UBE3A deficiency results in AS, UBE3A over-expression results in increased ASD risk. Increased density of dendritic spines with immature morphology has been reported in brains of ASD patients (*Hutsler and Zhang, 2010*; *Tang et al., 2014*). In our study, reducing p18 levels in WT mice resulted in similar changes in spine properties. It is therefore tempting to propose that UBE3A over-expression might induce ASD phenotypes, at least in part, by down-regulating p18 levels. It is also noteworthy that abnormal mTOR signaling has been implicated in a number of diseases. Therefore, results from our studies shed new light on a broad range of normal brain functions, and on several neurological/neuropsychiatric disorders, including AS.

# Materials and methods

**Key resources table**

| Reagent type (species) or resource | Designation | Source or reference | Identifiers | Additional information |
|---|---|---|---|---|
| Strain, strain background (*Mus musculus*) | B6.129S7-Ube3a^tm1Alb^/J | The Jackson Laboratory PMID: 9808466 | MGI: J:50811 | |
| Strain, strain background | Control shRNA Lentiviral Particles | Santa Cruz Biotechnology | sc-108080 | |
| Strain, strain background (*Mus musculus*) | P18 shRNA Lentiviral Particles | Santa Cruz Biotechnology | sc-108727-V | |
| Strain, strain background | copGFP Control Lentiviral Particles | Santa Cruz Biotechnology | sc-108084 | |
| Strain, strain background | Scrambled AAV9 siRNA Control Virus | Applied Biological Materials | iAAV01509 | |
| Strain, strain background (*Mus musculus*) | P18 AAV9 siRNA Pooled Virus | Applied Biological Materials | iAAV04811709 | |
| Strain, strain background | Scrambled AAV9 shRNA Control Virus | VectorBuilder | SP1001276 | Custom |
| Strain, strain background (*Mus musculus*) | P18 AAV9 shRNA Virus | VectorBuilder | SP1001275 | Custom; target sequence: 5'-CGTATGCCTATAGTGCA CTTT-3' |

*Continued on next page*

*Continued*

| Reagent type (species) or resource | Designation | Source or reference | Identifiers | Additional information |
|---|---|---|---|---|
| Genetic reagent (*Homo sapiens*) | SMARTpool ON-TARGETplus Human UBE3A siRNA | Dharmacon | L-005137–00 | |
| Genetic reagent | ON-TARGETplus Non-targeting Control Pool | Dharmacon | D-001810–10 | |
| Genetic reagent (*Mus musculus*) | SMARTpool Accell Mouse Ube3a siRNA | Dharmacon | E-047237–00 | |
| Genetic reagent | Accell Non-targeting Control Pool | Dharmacon | D-001910–10 | |
| Cell line (*Cercopithecus aethiops*) | COS-1 | ATCC | CRL-1650; RRID:CVCL_0223 | |
| Antibody | anti-UBE3A (mouse monoclonal) | Sigma-Aldrich | E8655; RRID:AB_261956 | (1:2000) |
| Antibody | anti-p18 (rabbit monoclonal) | Cell Signaling Technology | 8975; RRID:AB_10860252 | (1:1000) |
| Antibody | anti-p18 (rabbit polyclonal) | Sigma-Aldrich | HPA002997; RRID:AB_1845531 | (1:200) |
| Antibody | anti-p14 (rabbit monoclonal) | Cell Signaling Technology | 8145; RRID:AB_10971636 | (1:1000) |
| Antibody | anti-MP1 (rabbit monoclonal) | Cell Signaling Technology | 8168; RRID:AB_10949501 | (1:1000) |
| Antibody | anti-LAMTOR4 (rabbit monoclonal) | Cell Signaling Technology | 12284 | (1:500) |
| Antibody | anti-RagA (rabbit monoclonal) | Cell Signaling Technology | 4357; RRID:AB_10545136 | (1:1000) |
| Antibody | anti-RagB (rabbit monoclonal) | Cell Signaling Technology | 8150; RRID:AB_11178806 | (1:1000) |
| Antibody | anti-RagC (rabbit monoclonal) | Cell Signaling Technology | 5466; RRID:AB_10692651 | (1:1000) |
| Antibody | anti-ubiquitin (mouse monoclonal) | Enzo Life Sciences | BML-PW8810; RRID:AB_10541840 | (1:800) |
| Antibody | anti-Flag (mouse monoclonal) | Sigma-Aldrich | F1804; RRID:AB_262044 | (1:1000) |
| Antibody | anti-LAMP2 (rat monoclonal) | Abcam | ab13524; RRID:AB_2134736 | (1:200) |
| Antibody | anti-LAMP1 (mouse monoclonal) | Abcam | ab25630; RRID:AB_470708 | (1:20) |
| Antibody | anti-p-mTOR Ser2448 (rabbit polyclonal) | Cell Signaling Technology | 2971; RRID:AB_330970 | (1:1000) |
| Antibody | anti-mTOR (rabbit polyclonal) | Cell Signaling Technology | 2972; RRID:AB_330978 | (1:1000) |
| Antibody | anti-p-S6K1 Thr389 (rabbit polyclonal) | Cell Signaling Technology | 9205; RRID:AB_330944 | (1:1000) |
| Antibody | anti-S6K1 (rabbit polyclonal) | Cell Signaling Technology | 9202; RRID:AB_331676 | (1:1000) |
| Antibody | anti-p-S6 Ser240/244 (rabbit polyclonal) | Cell Signaling Technology | 2215; RRID:AB_331682 | (1:1000) |
| Antibody | anti-S6 (rabbit monoclonal) | Cell Signaling Technology | 2217; RRID:AB_331355 | (1:1000) |
| Antibody | anti-p-PKC (rabbit polyclonal) | Cell Signaling Technology | 9371; RRID:AB_2168219 | (1:1000) |
| Antibody | anti-PKCα (rabbit polyclonal) | Cell Signaling Technology | 2056; RRID:AB_2284227 | (1:1000) |

*Continued on next page*

*Continued*

| Reagent type (species) or resource | Designation | Source or reference | Identifiers | Additional information |
|---|---|---|---|---|
| Antibody | anti-p-4EBP1 Ser65 (rabbit polyclonal) | Cell Signaling Technology | 9451; RRID:AB_330947 | (1:1000) |
| Antibody | anti-4EBP1 (rabbit monoclonal) | Cell Signaling Technology | 9644; RRID:AB_2097841 | (1:1000) |
| Antibody | anti-p-AKT Ser473 (rabbit monoclonal) | Cell Signaling Technology | 4060; RRID:AB_2315049 | (1:1000) |
| Antibody | anti-AKT (rabbit polyclonal) | Cell Signaling Technology | 9272; RRID:AB_329827 | (1:1000) |
| Antibody | anti-Raptor (mouse monoclonal) | EMD Millipore | 05–1470; RRID:AB_10615925 | (1:500) |
| Antibody | anti-Rictor (rabbit polyclonal) | Bethyl Laboratories | A300-459A; RRID:AB_2179967 | (1:200) |
| Antibody | anti-NeuN (mouse monoclonal) | EMD Millipore | MAB377; RRID:AB_2298772 | (1:100) |
| Antibody | anti-PSD95 (mouse monoclonal) | Thermo Fisher Scientific | MA1-045; RRID:AB_325399 | (1:200) |
| Antibody | anti-Cathepsin B (mouse monoclonal) | EMD Millipore | IM27L; RRID:AB_2274848 | (1:400) |
| Antibody | anti-COXIV (rabbit monoclonal) | Cell Signaling Technology | 4850; RRID:AB_2085424 | (1:1000) |
| Antibody | anti-GFP (chicken polyclonal) | Thermo Fisher Scientific | A10262; RRID:AB_2534023 | (1:500) |
| Antibody | anti-GAPDH (mouse monoclonal) | EMD Millipore | MAB374; RRID:AB_2107445 | (1:1000) |
| Antibody | anti-β-actin (mouse monoclonal) | Sigma-Aldrich | A5441; RRID:AB_476744 | (1:10,000) |
| Antibody | Goat anti-rabbit IgG IRDye 680RD | LI-COR Biosciences | 926–68071 | (1:10,000) |
| Antibody | Goat anti-mouse IgG IRDye 800CW | LI-COR Biosciences | 926–32210 | (1:10,000) |
| Antibody | Alexa 488- secondaries | Molecular Probes | | (1:400) |
| Antibody | Alexa 594- or 633- secondaries | Molecular Probes | | (1:200) |
| Recombinant DNA reagent | HA-tagged wild-type Ube3a | Addgene PMID: 9497376 | 8648 | |
| Recombinant DNA reagent | HA-tagged Ube3a-C833A | Addgene PMID: 9497376 | 8649 | |
| Recombinant DNA reagent | HA-p18 | Addgene PMID: 22980980 | 42338 | |
| Recombinant DNA reagent | HA-p18G2A | Addgene PMID: 22980980 | 42327 | |
| Recombinant DNA reagent | Flag-p18 | Addgene PMID: 22980980 | 42331 | |
| Recombinant DNA reagent | Flag-p18ΔK | This paper | N/A | Custom Gene Synthesis from Integrated DNA Technologies (all lysine residues in p18 mutated into arginine) |
| Recombinant DNA reagent | Flag-p18K20R | This paper | N/A | Site-directed mutagenesis using a QuikChange II site-directed mutagenesis kit (Agilent). The mutation was confirmed by sequencing. |
| Recombinant DNA reagent | Flag-p18K31R | This paper | N/A | Same as above |

*Continued on next page*

*Continued*

| Reagent type (species) or resource | Designation | Source or reference | Identifiers | Additional information |
|---|---|---|---|---|
| Recombinant DNA reagent | Flag-p18K60R | This paper | N/A | Same as above |
| Recombinant DNA reagent | Flag-p18K103/104R | This paper | N/A | Same as above |
| Recombinant DNA reagent | Flag-p18K151R | This paper | N/A | Same as above |
| Recombinant DNA reagent | His-ubiquitin | Addgene PMID: 21183682 | 31815 | |
| Peptide, recombinant protein | Recombinant human p18 | CUSABIO | CSB-EP757561XBF | |
| Commercial assay or kit | E6AP (UBE3A) Ubiquitin Ligase Kit | Boston Biochem | K-240 | |
| Commercial assay or kit | FD Rapid GolgiStain Kit | FD Neurotechnologies | PK401 | |
| Chemical compound, drug | MG132 | EMD Millipore | 474790 | 10 μM |
| Chemical compound, drug | Bafilomycin A1 | Sigma-Aldrich | B1793 | 100 nM |
| Chemical compound, drug | MHY1485 | EMD Millipore | 500554 | 2 μM |
| Chemical compound, drug | Rhodamine Phalloidin | Molecular Probes | R415 | |
| Software, algorithm | ImageJ | https://imagej.nih.gov/ij/ | RRID:SCR_003070 | |
| Software, algorithm | Prism 6 | GraphPad Software | RRID:SCR_002798 | |

## Animals

Animal experiments were conducted in accordance with the principles and procedures of the National Institutes of Health Guide for the Care and Use of Laboratory Animals. All protocols were approved by the local Institutional Animal Care and Use Committee. Original Ube3a mutant (AS) mice were obtained from The Jackson Laboratory, strain B6.129S7-*Ube3a*^tm1Alb^/J, and a breeding colony was established, as previously described (*Baudry et al., 2012*). In all experiments male AS mice aged between 2 and 4 months were used. Control mice were age-matched, male, wild-type littermates. Mice, housed in groups of two to three per cage, were maintained on a 12 hr light/dark cycle with food and water ad libitum.

## Hippocampal neuronal cultures

For neuronal culture preparations, wild-type (WT) or *Ube3a*^m-/p+^ female and WT male mice were used for breeding. Hippocampal neurons were prepared from E18 mouse embryos as described (*Sun et al., 2015b*). Briefly, hippocampi were dissected and digested with papain (2 mg/ml, Sigma) for 30 min at 37°C. Dissociated cells were plated onto poly-D-lysine-coated six-well plates at a density of 6–10 $\times$ 10$^4$ cells/cm$^2$ or coverslips in 24-well plates at a density of 6–10 $\times$ 10$^3$ cells/cm$^2$ in Neurobasal medium (GIBCO) supplemented with 2% B27 (GIBCO) and 2 mM glutamine and kept at 37°C under 5% $CO_2$. Half of the culture medium was replaced with fresh culture medium at DIV4 and then every 7 d. Genotyping was carried out by polymerase chain reaction (PCR) of mouse tail DNA as described previously (*Baudry et al., 2012*).

## Cell lines

COS-1 cells (ATCC) were grown in DMEM supplemented with 10% (vol/vol) fetal bovine serum (FBS) (Invitrogen) and kept at 37°C under 5% $CO_2$. Low-passage cells were used, and the identity of cells has been authenticated by morphology check by microscope and growth curve analysis. No mycoplasma contamination was detected.

## Transfection and lentiviral infection

For transient expression of constructs, COS-1 cells were transfected with the respective constructs by lipofection (Lipofectamine 2000; Invitrogen) according to the manufacturer's instructions. Small interfering RNA (siRNA) transfections were also performed with Lipofectamine 2000. Cells were incubated with 10 or 20 nM SMARTpool siRNA duplexes against human *UBE3A*, or a scrambled duplex (Dharmacon) for 72 hr before downstream analysis.

Cultured hippocampal neurons from WT mice were infected with p18 shRNA (mouse) lentivirus (sc-108727-V, Santa Cruz Biotechnology) or scrambled shRNA lentivirus (sc-108080, Santa Cruz Biotechnology), and co-transfected with Accell Ube3a siRNA (GE Dharmacon) or Accell Non-targeting siRNA (GE Dharmacon) at DIV 4, and 24 hr after infection, two-thirds of the medium was replaced with fresh medium. Cultured neurons were used 3 d after infection.

Cultured WT and AS hippocampal neurons were infected with p18 shRNA (mouse) lentivirus (sc-108727-V, Santa Cruz Biotechnology) or scrambled shRNA lentivirus (sc-108080), together with copGFP control lentivirus (as an infection marker, sc-108084) at DIV 14, and 24 hr after infection, two-thirds of the medium was replaced with fresh medium. Neurons were analyzed 8 d after infection.

Cultured WT and AS hippocampal neurons were infected with p18 shRNA (mouse) AAV (custom, VectorBuilder) or scrambled shRNA AAV (custom, VectorBuilder) at DIV 7, and 24 hr after infection, two-thirds of the medium was replaced with fresh medium. Neurons were analyzed 14 d after infection.

## Antibodies, chemicals, and DNA constructs

Antibodies, chemicals, and plasmids used in this study are listed in *Table 1*.

All antibodies listed are validated for Western blot and/or immunohistochemistry by their respective sources. We also validated each IHC-recommended antibody following the Rimm Lab Algorithm (*Bordeaux et al., 2010*).

## P2/S2 fractionation, lysosomal fractionation, and western blot analysis

P2/S2 fractionation was performed according to published protocols (*Sun et al., 2015a*). Briefly, frozen hippocampus tissue was homogenized in ice-cold HEPES-buffered sucrose solution (0.32 M sucrose, 4 mM HEPES, pH 7.4) with protease inhibitors. Homogenates were centrifuged at 900 g for 10 min to remove large debris (P1). The supernatant (S1) was then centrifuged at 11,000 g for 20 min to obtain crude membrane (P2) and cytosolic (S2) fractions. P2 pellets were sonicated in RIPA buffer (10 mM Tris, pH 8, 140 mM NaCl, 1 mM EDTA, 0.5 mM EGTA, 1% NP-40, 0.5% sodium deoxycholate, and 0.1% SDS). For whole homogenates, tissue was homogenized in RIPA buffer. Protein concentrations were determined with a BCA protein assay kit (Pierce).

Lysosome-enriched fractions were prepared from cultured neurons or isolated hippocampus using the lysosome enrichment kit (Pierce). The purity of the fractions was assessed using antibodies against cathepsin B (lysosomes) and COXIV (mitochondria). Cultured hippocampal neurons from WT mice were transfected with Accell p18 siRNA or Accell Non-targeting siRNA (GE Dharmacon) at DIV 7, and were used 4 d after transfection. At least three independent experiments were performed.

Western blots were performed according to published protocols (*Sun et al., 2015b*). Briefly, samples were separated by SDS-PAGE and transferred onto a PVDF membrane (Millipore). After blocking with 3% BSA for 1 h, membranes were incubated with specific antibodies overnight at 4°C followed by incubation with secondary antibodies (IRDye secondary antibodies) for 2 hr at room temperature. Antibody binding was detected with the Odyssey family of imaging systems.

## Immunoprecipitation and denaturing immunoprecipitation

For immunoprecipitation, all procedures were carried out at 4°C. COS-1 cells transfected with the indicated cDNAs or cultured hippocampal neurons were lysed with lysis buffer (Tris-HCl 25 mM pH 7.4, NaCl 150 mM, 1 mM EDTA, 1% NP-40, 5% glycerol, and a protease inhibitor cocktail). After a brief centrifugation to remove insoluble material, the supernatant was precleared with an aliquot of agarose beads. For immunoprecipitation of Flag-p18 or Flag-p18ΔK in COS-1 cells, extracts were incubated overnight with anti-Flag agarose beads, washed with lysis buffer, followed by elution of bound proteins by heating at 95°C for 10 min in SDS-PAGE sample buffer. For immunoprecipitation

**Table 1.** Antibodies, chemicals, and plasmids used in this study

| Reagent or resource | Source | Identifier |
| --- | --- | --- |
| Antibodies | | |
| Mouse monoclonal anti-UBE3A (clone E6AP-330) | Sigma-Aldrich | Cat#E8655 |
| Rabbit monoclonal anti-p18 | Cell Signaling Technology | Cat#8975 |
| Rabbit polyclonal anti-p18 | Sigma-Aldrich | Cat#HPA002997 |
| Rabbit monoclonal anti-p14 | Cell Signaling Technology | Cat#8145 |
| Rabbit monoclonal anti-MP1 | Cell Signaling Technology | Cat#8168 |
| Rabbit monoclonal anti-LAMTOR4 | Cell Signaling Technology | Cat#12284 |
| Rabbit monoclonal anti-RagA | Cell Signaling Technology | Cat#4357 |
| Rabbit monoclonal anti-RagB | Cell Signaling Technology | Cat#8150 |
| Rabbit monoclonal anti-RagC | Cell Signaling Technology | Cat#5466 |
| Mouse monoclonal anti-ubiquitin | Enzo Life Sciences | Cat#BML-PW8810 |
| Mouse monoclonal anti-Flag | Sigma-Aldrich | Cat#F1804 |
| Rat monoclonal anti-LAMP2 | Abcam | Cat#ab13524 |
| Mouse monoclonal anti-LAMP1 | Abcam | Cat#ab25630 |
| Rabbit polyclonal anti-p-mTOR Ser2448 | Cell Signaling Technology | Cat#2971 |
| Rabbit polyclonal anti-mTOR | Cell Signaling Technology | Cat#2972 |
| Rabbit polyclonal anti-p-S6K1 Thr389 | Cell Signaling Technology | Cat#9205 |
| Rabbit polyclonal anti-S6K1 | Cell Signaling Technology | Cat#9202 |
| Rabbit polyclonal anti-p-S6 Ser240/244 | Cell Signaling Technology | Cat#2215 |
| Rabbit monoclonal anti-S6 | Cell Signaling Technology | Cat#2217 |
| Rabbit polyclonal anti-p-PKC | Cell Signaling Technology | Cat#9371 |
| Rabbit polyclonal anti-PKCα | Cell Signaling Technology | Cat#2056 |
| Rabbit polyclonal anti-p-4EBP1 Ser65 | Cell Signaling Technology | Cat#9451 |
| Rabbit monoclonal anti-4EBP1 | Cell Signaling Technology | Cat#9644 |
| Rabbit monoclonal anti-p-AKT Ser473 | Cell Signaling Technology | Cat#4060 |
| Rabbit polyclonal anti-AKT | Cell Signaling Technology | Cat#9272 |
| Mouse monoclonal anti-Raptor (clone 1H6.2) | EMD Millipore | Cat#05–1470 |
| Rabbit polyclonal anti-Rictor | Bethyl Laboratories | Cat#A300-459A |
| Mouse monoclonal anti-NeuN (clone A60) | EMD Millipore | Cat#MAB377 |
| Mouse monoclonal anti-PSD95 (clone 6G6-1C9) | Thermo Fisher Scientific | Cat#MA1-045 |
| Mouse monoclonal anti-Cathepsin B (clone CA10) | EMD Millipore | Cat#IM27L |
| Rabbit monoclonal anti-COXIV | Cell Signaling Technology | Cat#4850 |
| Chicken polyclonal anti-GFP | Thermo Fisher Scientific | Cat#A10262 |
| Mouse monoclonal anti-GAPDH (clone 6C5) | EMD Millipore | Cat#MAB374 |
| Mouse monoclonal anti-β-actin (clone AC-15) | Sigma-Aldrich | Cat#A5441 |
| Goat anti-rabbit IgG IRDye 680RD | LI-COR Biosciences | Cat#926–68071 |
| Goat anti-mouse IgG IRDye 800CW | LI-COR Biosciences | Cat#926–32210 |
| Goat anti-mouse IgG AlexaFluor 488 | Invitrogen | Cat#A-11029 |
| Goat anti-rabbit IgG AlexaFluor 594 | Invitrogen | Cat#A-11037 |
| Goat anti-rat IgG AlexaFluor 488 | Invitrogen | Cat#A-11006 |
| Goat anti-rat IgG AlexaFluor 594 | Invitrogen | Cat#A-21209 |
| Goat anti-rabbit IgG AlexaFluor 488 | Invitrogen | Cat#A-11008 |
| Goat anti-chicken IgY AlexaFluor 488 | Invitrogen | Cat#A-11039 |
| Goat anti-mouse IgG AlexaFluor 633 | Invitrogen | Cat#A-21052 |

*Table 1 continued*

| Reagent or resource | Source | Identifier |
|---|---|---|
| Goat anti-rabbit IgG AlexaFluor 633 | Invitrogen | Cat#A-21070 |
| Chemicals | | |
| MG132 | EMD Millipore | Cat#474790 |
| Bafilomycin A1 | Sigma-Aldrich | Cat#B1793 |
| MHY1485 | EMD Millipore | Cat#500554 |
| Recombinant DNA | | |
| HA-tagged wild-type Ube3a | (*Talis et al., 1998*) | Addgene #8648 |
| HA-tagged Ube3a-C833A | (*Talis et al., 1998*) | Addgene #8649 |
| HA-p18 | (*Bar-Peled et al., 2012*) | Addgene #42338 |
| HA-p18G2A | (*Bar-Peled et al., 2012*) | Addgene #42327 |
| Flag-p18 | (*Bar-Peled et al., 2012*) | Addgene #42331 |
| Flag-p18ΔK | This paper | N/A |
| Flag-p18K20R | This paper | N/A |
| Flag-p18K31R | This paper | N/A |
| Flag-p18K60R | This paper | N/A |
| Flag-p18K103/104R | This paper | N/A |
| Flag-p18K151R | This paper | N/A |
| His-ubiquitin | (*Young et al., 2011*) | Addgene #31815 |

DOI: https://doi.org/10.7554/eLife.37993.027

of p18 in hippocampal neurons, extracts were incubated with anti-p18 antibodies overnight and immunoprecipitates were collected with protein A/G Agarose. For immunoprecipitation of RagA in mouse hippocampus, rabbit anti-RagA antibodies were incubated with hippocampal lysates and precipitated with Protein A/G-conjugated beads. Inputs and precipitates were resolved by SDS-PAGE and analyzed by Western blotting. All studies were performed in three to five independent experiments.

For immunoprecipitation of ubiquitin from hippocampal crude membrane fractions under denaturing conditions, P2 pellets were resuspended and heated in denaturing lysis buffer (1% SDS, 50 mM Tris, pH 7.4, 5 mM EDTA, 10 mM DTT, 1 mM PMSF, 2 µg/ml leupeptin, 15 U/ml DNase I) and diluted in nine volumes of ice-cold non-denaturing lysis buffer (1% Triton X-100, 50 mM Tris, pH 7.4, 300 mM NaCl, 5 mM EDTA, 10 mM iodoacetamide, 1 mM PMSF, 2 µg/ml leupeptin). Lysates were centrifuged at 16,000 g for 30 min at 4°C and cleared with protein A/G Agarose beads. Pre-cleared lysates were then incubated with anti-ubiquitin antibodies coupled to protein A/G Agarose beads overnight at 4°C, followed by four washes with ice-cold wash buffer (0.1% Triton X-100, 50 mM Tris, pH 7.4, 300 mM NaCl, 5 mM EDTA) and elution in 2 x SDS sample buffer. Immunoprecipitated proteins were resolved by SDS-PAGE followed by Western blot analysis with specific antibodies against p18 and ubiquitin. Relative p18 ubiquitination refers to the ratio of ubiquitinated p18 over total p18, and was normalized to the average value of the WT group. At least three independent experiments were performed.

## In vitro ubiquitination assay

His-p18 proteins were purchased from CUSABIO (Wuhan, China). For in vitro ubiquitination experiments, we used the E6AP (UBE3A) Ubiquitin Ligase Kit (Boston Biochem), following the manufacturer's instructions. Briefly, purified His-p18 proteins were incubated for 90 min at 37°C under constant shaking with E1 enzyme, E2 enzyme (UBE2L3), $His_6$-E6AP, ubiquitin, $Mg^{2+}$-ATP, and reaction buffer. The reaction was terminated by the addition of SDS sample buffer, and samples were boiled, and proteins separated with 14% SDS-PAGE. Blots were probed with p18, ubiquitin, and His antibodies. At least three independent experiments were performed.

## His-ubiquitin pull-down assay

COS-1 cells in 60 mm dishes were transfected with 2.5 µg His-ubiquitin, 2.5 µg HA-p18 or HA-p18G2A, and 5 µg HA-Ube3a or HA-Ube3a C833A constructs in the indicated combinations. Ube3a siRNA-treated COS-1 cells were transfected with 2.5 µg Flag-p18 or p18ΔK and 2.5 µg His-ubiquitin 48 hr after siRNA treatment. Twenty-four hours after transfection, cells were lysed, and His-ubiquitin-conjugated proteins were purified as described (*Sun et al., 2015b*). Briefly, cells were harvested in 1 ml of ice-cold phosphate-buffered saline, and the cell suspension was divided into two parts; 100 µl were lysed using 1 x SDS-PAGE sample loading buffer containing 10% DTT, and 900 µl were lysed in Buffer A (6 M guanidine HCl, 0.1 M $Na_2HPO_4/NaH_2PO_4$, 0.5 M NaCl, 10 mM imidazole, 0.1% Nonidet P-40, and 5% glycerol, pH 8.0) and sonicated. The guanidine lysates were incubated with 30 µl of equilibrated Talon resin at 4°C for 4 hr to bind His-tagged ubiquitinated proteins. Beads were then washed one time with Buffer A, followed by four washes with Buffer B (8 M urea, 0.1 M $Na_2HPO_4/NaH_2PO_4$, 0.5 M NaCl, 20 mM imidazole, 0.1% Nonidet P-40, and 5% glycerol, pH 8.0). The protein conjugates were eluted in 30 µl of 2 X laemmli/imidazole (200 mM imidazole) and boiled. Eluates were analyzed by Western blotting using either p18 or ubiquitin antibody. At least three independent experiments were performed.

## Acute hippocampal slice preparation

Adult male mice (2–4 month-old) were anesthetized with gaseous isoflurane and decapitated. Brains were quickly removed and transferred to oxygenated, ice-cold cutting medium (in mM): 124 NaCl, 26 $NaHCO_3$, 10 glucose, 3 KCl, 1.25 $KH_2PO_4$, 5 $MgSO_4$, and 3.4 $CaCl_2$. Hippocampal transversal slices (400 µm-thick) were prepared using a McIlwain-type tissue chopper and transferred to i) an interface recording chamber and exposed to a warm, humidified atmosphere of 95% $O_2$/5% $CO_2$ and continuously perfused with oxygenated and preheated (33 ± 0.5°C) artificial cerebrospinal fluid (aCSF) containing (in mM): 110 NaCl, 5 KCl, 2.5 $CaCl_2$, 1.5 $MgSO_4$, 1.24 $KH_2PO_4$, 10 D-glucose, 27.4 $NaHCO_3$, perfused at 1.4 ml/min (electrophysiology); or ii) a recovery chamber with a modified aCSF medium, containing (in mM): 124 NaCl, 2.5 KCl, 2.5 $CaCl_2$, 1.5 $MgSO_4$, 1.25 $NaH_2PO_4$, 24 $NaHCO_3$, 10 D-glucose, and saturated with 95% $O_2$/5% $CO_2$ for 1 hr at 37°C (biochemical assays).

## Electrophysiology

After 1.5 hr incubation at 33 ± 0.5°C in the recording chamber, a single glass pipette filled with 2 M NaCl was used to record field EPSPs (fEPSPs) elicited by stimulation of the Schaffer collateral pathway with twisted nichrome wires (single bare wire diameter, 50 µm) placed in CA1 stratum radiatum. Stimulation pulses were generated using a Multichannel Systems Model STG4002 Stimulator (Reutlingen, Germany). Responses were recorded through a differential amplifier (DAM 50, World Precision Instruments, USA) with a 10 kHz high-pass and 0.1 Hz low-pass filter. Before each experiment, the input/output (I/O) relation was examined by varying the intensity of the stimulation. Paired-pulse facilitation was tested at 20–300 ms interval. Long-term potentiation (LTP) was induced using theta burst stimulation (10 bursts at 5 Hz, each burst consisting of four pulses at 100 Hz, with a pulse duration of 0.2 ms). For LTP and paired-pulse facilitation experiments, the stimulation intensity was regulated to a current which elicited a 40% of maximal response. Data were collected and digitized by Clampex, and the slope of fEPSP was analyzed. MHY1485 (2 µM) was applied to slices for 60 min before theta-burst stimulation (TBS). Some of the slices were processed for Western blots. All data are expressed as means ± SEM, and statistical significance of differences between means was calculated with appropriate statistical tests as indicated in figure legends.

Whole-cell patch-clamp recording was performed as previously described (*Vogel-Ciernia et al., 2013*). Briefly, hippocampal slices were prepared on the horizontal plane at a thickness of 370 µm from 2- to 4-month-old male mice with a Leica vibrating tissue slicer (Model: VT1000S). Slices were placed in a submerged recording chamber and continuously perfused at 2–3 mL/min with oxygenated (95% $O_2$/5% $CO_2$) aCSF at 32°C. Whole-cell recordings (Axopatch 200A amplifier: Molecular Devices) were made with 4–7 MΩ recording pipettes filled with a solution containing (in mM): 130 $CsMeSO_4$, 10 CsCl, 8 NaCl, 10 HEPES, 0.2 EGTA, 5 QX-314, 2 Mg-ATP, 0.3 Na-GTP. Osmolarity was adjusted to 290–295 mOsm and pH 7.4. Spontaneous mEPSCs were recorded at a holding potential of –70 mV in the presence of tetrodotoxin (1 µM) and picrotoxin (50 µM). Data were filtered at 2 kHz, digitized at 1–5 kHz, stored on a computer, and analyzed off-line using Mini Analysis Program

(Synaptosoft), Origin (OriginLab) and pCLAMP 7 (Molecular Devices) software. Statistical significance was determined by pooling events from cells of the same genotype and running a Student's t-test on the pooled data. p<0.05 was considered statistically significant.

## Immunofluorescence

Cultured hippocampal neurons were fixed in 2% paraformaldehyde (PFA)/10% sucrose for 15 min at 37°C, transferred to 0.05% Triton X-100/PBS for 5 min at 4°C, and then 0.02% Tween-20/PBS for 2 min at 4°C. Coverslips were washed twice with ice cold PBS and incubated 1 hr in 3% BSA/PBS at room temperature. For staining of F-actin, Rhodamine-Phalloidin (Invitrogen) was incubated in 1% BSA/PBS overnight at 4°C. For staining of p18, LAMTOR4, RagA, and LAMP2, cells were incubated with rabbit anti-p18 (1:200, Sigma), rabbit anti-LAMTOR4 (1:500, CST), rabbit anti-RagA (1:100, CST), rat anti-LAMP2 (1:200, Abcam) respectively in 3% BSA/PBS overnight at 4°C. Coverslips were then washed twice with ice cold PBS for 10 min each and then incubated with secondary antibodies (Alexa Fluor-594 anti-rabbit, 1:200; Alexa Fluor-594 anti-rat, 1:200; and Alexa Fluor-633 anti-mouse, 1:200) for 2 hr at room temperature. Coverslips were then washed four times with ice cold PBS for 10 min each, and mounted on glass slides using VECTASHIELD mounting medium with DAPI (Vector Laboratories). Images were acquired using a Zeiss LSM 880 confocal laser-scanning microscope. The staining was visualized in GFP-expressed neurons. Mean fluorescence intensity (MFI) was calculated over a specific region of interest, and background staining of the sections was measured and subtracted from the total signal to obtain the specific signal.

Hippocampal slices were collected 40 min after TBS and fixed in 4% PFA for 1 hr and cryoprotected in 30% sucrose for 1 hr at 4°C, and sectioned on a freezing microtome at 20 μm. Sections were blocked in 0.1 M PBS containing 5% goat serum and 0.3% Triton X-100, and then incubated in primary antibody mixture including chicken anti-GFP (1:500) and rabbit anti-p18 (1:200, Sigma) in 0.1 M PBS containing 1% BSA and 0.3% Triton X-100 overnight at 4°C. Sections were washed three times (10 min each) in PBS and incubated in Alexa Fluor 488 goat anti-chicken IgG and Alexa Fluor 594 goat anti-rabbit IgG for 2 hr at room temperature. All images were taken in CA1 stratum radiatum between the stimulating and recording electrodes. The threshold for the GFP fluorescence was set to make sure that the control slices from naive mice or mice with AAV infection but without GFP reporter were considered GFP-negative.

For immunofluorescence with brain tissue section, deeply anesthetized animals were perfused and brains were post-fixed in 4% PFA overnight followed by sequential immersion in 15% and 30% sucrose for cryoprotection. Brains were then sectioned (20 μm) and stained as described above. The following primary antibodies were used: p18 (1:200, Sigma), LAMTOR4 (1:500, CST), RagA (1:100, CST), p14 (1:100, CST), MP1 (1:100, CST), RagB (1:100, CST) mTOR (1:100, CST), p-mTOR (1:100, CST), LAMP2 (1:200, Abcam), and PSD95 (1:200, Thermo). The hippocampal CA1 pyramidal cell soma and apical dendrites were randomly selected for colocalization analysis by Manders' coefficients. The apical dendrites in hippocampal CA1 stratum radiatum were also randomly selected for puncta analysis. The puncta number of p18/PSD95 was quantified and the percentage of p18 and PSD95 dually stained synapses was also analyzed.

## Intrahippocampal AAV injection

We employ a dual convergent promoter system (U6 and H1 promoters) in which the sense and antisense strands of the siRNA are expressed by two different promoters rather than in a hairpin loop to avoid any possible recombination events. Stereotaxic AAV injection into CA1 region of the hippocampus was performed in 8-week-old mice. Animals were allocated into the experimental/control groups in a randomized manner. Under isoflurane anesthesia, AAV p18 siRNA or AAV scrambled siRNA constructs in 2 μl solution were injected bilaterally into CA1 regions at two sites: 1.94 mm posterior to bregma, 1.4 mm lateral to the midline and 1.35 mm below the dura and 2.2 mm posterior to bregma, 1.8 mm lateral to the midline and 1.5 mm below the dura. The solution was slowly injected over 30 min and the needle was left in place for an additional 10 min. The needle was then slowly withdrawn and the incision closed. AAV-injected mice were used for experiments after 4 weeks, a period determined in pilot studies to be necessary for sufficient expression of viral mediated gene expression.

## Image analysis and quantification

Images were acquired using a Nikon C1 or a Zeiss LSM 880 with Airyscan confocal laser-scanning microscope with a 60 X objective. Images for all groups in a particular experiment were obtained using identical acquisition parameters and analyzed using ImageJ software (NIH). All immunostaining studies were performed in three to five independent experiments. In all cases the experimenter was blinded regarding the identity of the transfected constructs and the genotypes during acquisition and analysis.

## Dendritic spine analysis

Four weeks after AAV injection, mice were deeply anesthetized using gaseous isoflurane and then decapitated. The brain was rapidly removed and Golgi impregnation was performed according to our published protocol (*Sun et al., 2016*) and outlined in the FD Rapid GolgiStain Kit (FD Neurotechnologies, Ellicott, MD). The number of spines located on randomly selected dendritic branches was counted manually by an investigator blinded to genotype and injection. Spine density was calculated by dividing the number of spines on a segment by the length of the segment and was expressed as the number of spines per μm of dendrite. Spine types were determined on the basis of the ratio of the width of the spine head to the length of the spine neck and classified as previously described (*Risher et al., 2015*). Five to seven dendritic branches between 10 and 20 μm in length were analyzed and averaged to provide a section mean.

## Fear conditioning

AS mice and their WT littermates were randomly assigned to either control or p18 siRNA groups and blinded to the examiner. Four weeks after AAV injection, mice were placed in the fear-conditioning chamber (H10-11M-TC, Coulbourn Instruments). The conditioning chamber was cleaned with 10% ethanol to provide a background odor. A ventilation fan provided a background noise at ~55 dB. After a 2 min exploration period, three tone-footshock pairings separated by 1 min intervals were delivered. The 85 dB 2 kHz tone lasted 30 s and co-terminated with a footshock of 0.75 mA and 2 s. Mice remained in the training chamber for another 30 s before being returned to home cages. Context test was performed 1 day after training in the original conditioning chamber with 5 min recording. On day 3, animals were subjected to cue/tone test in a modified chamber with different texture and color, odor, background noise, and lighting. After 5 min recording, mice were exposed to a tone (85 dB, 2 kHz) for 1 min. Mouse behavior was recorded with the Freezeframe software and data were analyzed using the Freezeview software (Coulbourn Instruments). Motionless bouts lasting more than 1 s were considered as freezing. The percent of time animal froze was calculated and group means with SEM were analyzed.

## Statistical analysis

Error bars indicate standard error of the mean. To compute p values, unpaired Student's t-test, and one- or two-way ANOVA with Tukey's post-test were used (GraphPad Prism 6), as indicated in figure legends. The level of statistical significance was set at $p < 0.05$.

## Additional information

### Funding

| Funder | Grant reference number | Author |
|---|---|---|
| National Institute of Neurological Disorders and Stroke | R01NS104078 | Michel Baudry |
| National Institute of Mental Health | R15MH101703 | Xiaoning Bi |

The funders had no role in study design, data collection and interpretation, or the decision to submit the work for publication.

## Author contributions
Jiandong Sun, Conceptualization, Data curation, Formal analysis, Validation, Investigation, Visualization, Methodology, Writing—original draft, Writing—review and editing; Yan Liu, Data curation, Formal analysis, Validation, Investigation, Methodology; Yousheng Jia, Data curation, Formal analysis, Validation, Investigation; Xiaoning Hao, Data curation, Validation, Investigation, Visualization; Wei ju Lin, Data curation, Formal analysis, Validation, Visualization; Jennifer Tran, Data curation, Validation, Visualization, Methodology; Gary Lynch, Formal analysis, Validation, Writing—review and editing; Michel Baudry, Conceptualization, Formal analysis, Funding acquisition, Validation, Investigation, Writing—original draft, Project administration, Writing—review and editing; Xiaoning Bi, Conceptualization, Resources, Formal analysis, Supervision, Funding acquisition, Validation, Investigation, Writing—original draft, Project administration, Writing—review and editing

## Author ORCIDs
Jiandong Sun (iD) http://orcid.org/0000-0003-4270-8704
Wei ju Lin (iD) https://orcid.org/0000-0003-4578-4120
Xiaoning Bi (iD) http://orcid.org/0000-0001-7449-7003

## Ethics
Animal experimentation: Animal studies were performed in strict accordance with the recommendations in the Guide for the Care and Use of Laboratory Animals of the National Institutes of Health and were according to protocols (#R17-022, #R18-007) approved by institutional animal care and use committee (IACUC) of Western University of Health Sciences.

## Decision letter and Author response
Decision letter https://doi.org/10.7554/eLife.37993.030
Author response https://doi.org/10.7554/eLife.37993.031

# Additional files

## Supplementary files
• Transparent reporting form
DOI: https://doi.org/10.7554/eLife.37993.028

## Data availability
All data generated or analyzed during this study are included in the manuscript and supporting files. Source data files have been provided for all figures.

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
