## [Decision Letter]

[Editors’ note: a previous version of this study was rejected after peer review, but the authors submitted for reconsideration. The first decision letter after peer review is shown below.]

Thank you for submitting your work entitled "UBE3A-mediated p18/LAMTOR1 ubiquitination and degradation regulate mTORC1 activity and synaptic plasticity" for consideration by *eLife*. Your article has been reviewed by three peer reviewers, one of whom is a member of our Board of Reviewing Editors, and the evaluation has been overseen by a Senior Editor. The reviewers have opted to remain anonymous.

Our decision has been reached after consultation between the reviewers. Based on these discussions and the individual reviews below, we regret to inform you that your work will not be considered further for publication in *eLife*.

The reviewers appreciated the conceptual premise of the manuscript, as you may have uncovered a previously unappreciated mechanism. However, there are experimental and conceptual (i. e. how the molecular process is linked to the LTP phenotype) issues that were raised by all reviewers. Thus, there is a significant amount of work that is required be performed to render this work acceptable for publication in *eLife*. For example, you should examine Arc more closely rather than bringing it up in the Discussion.

*Reviewer #1:*

Sun et al. investigated a novel mechanism regulating mTORC1 activity via Ube3a-mediated ubiquitination and degradation of p18 (LAMTOR1) and implicate this regulatory axis in proper hippocampal morphology and function. Previous studies have demonstrated the physiological role of lysosomal recruitment and activation of mTORC1 via the Rag-Ragulator complex. The upstream mechanisms, however, remain poorly defined especially in the brain. To this end, the authors first demonstrate that UBE3A, an E3 ligase, interacts with and ubiquitinates p18 in vitro and in hippocampal cells. Loss of UBE3A (as in the Angelman syndrome (AS) mouse model) or inhibiting the proteasome causes aberrant lysosomal accumulation of hippocampal p18. This effect coincides with increased lysosomal localization of RagA and mTOR and elevated mTORC1 activity. Importantly, reducing elevated levels of p18 in the AS mouse brain rescues impaired spine maturation and LTP.

1) The authors claim that AAV siRNA-mediated p18 downregulation in AS mice restored LTP to wild type control levels, yet the same treatment in wild-type mice engendered a significant reduction in LTP. These findings are hard to reconcile, since mTOR activity is reduced by knockdown of p18 in both wild type and AS mice neurons. An explanation for this discrepancy is required.

2) Figure 2C. The amount of GFP in the shScrambled versus shP18 groups appears different. Is this due to different image exposure times or different transfection efficiency? This is problematic vis-à-vis the accuracy quantification of p18 in the treatment group.

3) Figure 4H. The authors claim that in Ube3a-deficient neurons, excessive mTORC1 is localized to the lysosome resulting in enhanced activity. Is it possible that both mTORC1 and mTORC2 are being localized to the lysosome under these conditions? How can the authors distinguish between these complexes given their data?

4) Figure 5D. Treatment of cultured hippocampal neurons with shP18 in wild-type mice causes an increase in the number of spines but in Figure 7B, wild type mice treated with siP18 show a reduction in the number of mature spines. A rationale for this discrepancy should be included in the Discussion.

5) Some of the immunofluorescent quantification data (Figure 2F and Figure 4B) are not accompanied by a representative image. These should be included with the corresponding figure.

6) Some of the Western blots represent composite gels. Thus, the β-actin and GAPDH loading control are not valid.

7) Figure 1E and Figure 3C are missing input lanes.

8) For some of the normalized data, error bars are missing for control groups.

9) In Figure 3E, a counter stain such as DAPI should be included for visual confirmation of consistent exposure between groups.

*Reviewer #2:*

The manuscript by Sun et al. describes the novel interaction between Ube3a and p18 that results in regulation of neuronal mTORC1 signaling. This group has previously reported that mTORC1 activity is increased in the cerebellum and hippocampus of Ube3a knockout mice and that Inhibition of mTORC1 with rapamycin improved motor and learning phenotypes these mice. How Ube3a loss results in mTORC1 overactivation has been a mystery. This current manuscript article provides evidence that this is mediated through p18 (LAMTOR1) ubiqiutination. Loss of Ube3a ubiquitination of p18 (LAMTOR1) results in defects associated with synaptic plasticity and dendritic spine development. The authors suggest that this is related to increased lysosomal localization of p18 and resulting increased mTORC1 activation. Taken together, this study provides a novel mechanism of regulation of p18 by Ube3a ubiquitination. The authors should be commended on a comprehensive evaluation of the implications of this finding. However, there are a few questions that remain to be answered as described below.

1) Figure 1—figure supplement 1A: It seems like p18 protein levels are dramatically reduced with His-ubiquitin expression, even in the absence of Ube3a expression. How do the authors interpret this?

2) Figure 1E and Figure 3C: y-axes are labeled as relative p18 ubiquitination. How is that calculated? Is it the ratio of ubiquitinated p18 over total p18?

3) Figure 3D: p18 protein levels are increase with MG132 treatment even in Ube3a KO mice. What is the explanation for this increase?

4) In the Results (subsection, “4. Increased p18 levels in AS mice are associated with increased lysosomal localization of the Ragulator-Rag complex and mTOR”), the authors study the localization of "p-mTOR, the active isoform of mTOR." I believe they are using Ser2448 phospho-mTOR antibody for that purpose. There is some controversy about the role of Ser2448 phosphorylation on mTOR activity. Substitution of Ser2448 with alanine does not affect mTOR activity (Sekulic et al., 2000). If there is good evidence that this phospho-specific antibody correlates with mTOR activity, it should be included in this section.

5) S6 ribosomal protein can get phosphorylated at Ser235/236 by S6Ks; however, S6K1/2 are not the only enzymes that are able to phosphorylate these sites. The contribution of S6K1/2 in S6 phosphorylation was addressed using S6K1/S6K2 double knock-out animals, which were found to display no phosphorylation of S6 at Ser240/244, but persistent phosphorylation at Ser235/236 (Pende et al., Mol. Cell., 2004). Kinases other than S6K can phosphorylate Ser235/236. They should investigate whether phosphorylation of S6 at Ser240/244 is affected by p18.

6) Quantification method for both imaging and western blotting should be further described:

a) Specifically, for imaging quantification how was mean fluorescence intensity calculated, over the entire image field or a specific region of interest?

b) I cannot figure out how Figure 2F was generated. I assume the authors combined quantification of different double-staining experiments. This should be clarified. It should also be clarified what the n refers to in the figure legend regarding Figure 2F.

c) The ideal figure to include in Figure 2 would be one where are representative images of p18 staining and LAMP staining simultaneously of neurons treated with p18 shRNA or control shRNA. I could not find such a figure in the manuscript.

7) The authors show that knockdown of p18 results in loss of lysosomal localization of the Ragulator complex and Rag in hippocampal neurons by immunofluorescence. This is an important finding. Another way to make this assertion more robust would be to prepare lysosomal fractions and demonstrate these findings biochemically.

8) Figure 4: Do AS hippocampi have more lysosomes? How do we know that these lysosomes are neuronal vs. glia? There is no NeuN or other stain to determine the cell type that are being imaged.

9) Figures 5B: Why is p-PKC/GAPDH ratio quantified? The quantification should be done showing the level of the phospho-protein relative to its respective total protein (phospho-protein/total protein) after their respective normalization to the housekeeping protein.

10) Figure 6—figure supplement 1 – only total PKC levels were reported. Is that meant to show phospho-PKC?

11) Along the same lines, Figure 5 p-mTOR and p-S6 are modest and probably the least convincing data of the paper. It would be good to include other direct targets substrates of mTOR as markers of mTOR activation (e.g. p70/S6K and 4EBP1).

12) A previous paper by Tang et al. (2014, Neuron, PMID: 25155956) argues that mTORC1 over-activation results in increased spines. How do the authors reconcile this report with their findings?

13) Discussion: The authors propose that over-expression of Arc may be mechanistic link between Ube3a deficiency and LTP deficit. This argument has some holes. In their previous publication for example, these authors showed that rapamycin treatment decreased Arc levels in WT hippocampal slices, but did not affect LTP or spine morphology in WT mice (Sun et al., 2016). It seems like it would be easy to check Arc protein levels in their system to see if Arc is affected by p18 manipulations.

*Reviewer #3:*

Title: UBE3A-mediated p18/LAMTOR1 ubiquitination and degradation regulate mTORC1 3 activity and synaptic plasticity. The authors demonstrated the UBE3A ligase interacts and regulates p18 levels in primary hippocampal neurons and mice hippocampus.

The authors showed that Ube3a deficiency in AS mice increases levels of p18 and enhance lysosomal localization of p18 with other members of Regulator-Rag complex, promoting mTORC1 activation, spine morphology and changes in LTP. By down-regulating p18 levels, the authors reduced mTORC1 activation, improved LTP phenotype and dendritic spine maturation.

Method.

1) The manuscript is built on blots and IHC, please provide all source data for all experiments (blots) and add validation for Ab use (IHC). This is critical for any interpretation of the data. Assuming the authors provide the source data and control the first part of the manuscript is solid and important.

2) I did not see any description of the vectors? Which promoter they used? General neuronal specific? This can change the story accordingly (specific neuronal phenotype versus general).

The LTP phenotype is interesting however, the authors do not explain the rescue mechanistically, if it’s Arc dependent – show it, if there is an effect on protein synthesis – show it.

In order to increase the manuscript impact, rescue behavioral phenotype using similar manipulation to CA1 region of the hippocampus is needed. The authors can use MWM and/or contextual fear conditioning.

[Editors’ note: what now follows is the decision letter after the authors submitted for further consideration.]

Thank you for resubmitting your work entitled "UBE3A-mediated p18/LAMTOR1 ubiquitination and degradation regulate mTORC1 activity and synaptic plasticity" for further consideration at *eLife*. Your revised article has been favorably evaluated by Huda Zoghbi (Senior Editor), a Reviewing Editor, and two reviewers.

The manuscript has been improved but there are some remaining issues that need to be addressed before acceptance, as outlined below:

The revised manuscript by Sun et al. describes the novel interaction between Ube3a and p18 that results in regulation of neuronal mTORC1 signaling. The authors have addressed many of the critical questions and provided the source data to better evaluate the manuscript. In general, this study provides a novel mechanism of regulation of p18 by Ube3a ubiquitination. There is one major point below that should be addressed to strengthen the manuscript.

The direct link the authors make between LTP and synaptic structure they measured is not that obvious. Kaphzan et al. reported changes in IPSP, but not EPASP in CA1 region in AS mice:

Reversal of impaired hippocampal long-term potentiation and contextual fear memory deficits in Angelman syndrome model mice by evbB inhibitors.

Kaphzan H, Hernandez P, Jung JI, Cowansage KK, Deinhardt K, Chao MV, Abel T, Klann E.

Biol Psychiatry 2012 Mar 03;72(3):182-90. doi: 10.1016/j.biopsych.2012.01.021. Can the authors explain this discrepancy?

---

## [Author Response]

[Editors’ note: the author responses to the first round of peer review follow.]

The reviewers had two major common issues: (i) the lack of data showing the rescue of the behavioral phenotype using the similar manipulation (injection of siRNA against p18) to the CA1 region of the hippocampus we used for the rescue of the LTP, and (ii) the lack of data examining the effects of this manipulations on Arc levels. As we were anticipating these questions, we have performed these experiments, and the new results address these two points and further support our argument regarding the novel function of p18 in synaptic plasticity and learning and memory. We believe that these new data should satisfy the reviewers, as well as the Editors, and make the paper now suitable for publication in *eLife*.

Reviewer #1:[…] 1) The authors claim that AAV siRNA-mediated p18 downregulation in AS mice restored LTP to wild type control levels, yet the same treatment in wild-type mice engendered a significant reduction in LTP. These findings are hard to reconcile, since mTOR activity is reduced by knockdown of p18 in both wild type and AS mice neurons. An explanation for this discrepancy is required.

That the same p18 knockdown (KD) treatment in AS and WT mice resulted in different outcomes is most likely due to different basal p18 levels and mTORC1 activity in the two genotypes. AS mice have higher than normal basal levels of p18 and mTORC1 activity. As shown in Figure 6, p18 KD in WT mice induced a larger reduction in p18 levels than in AS mice. Correspondingly, p18 KD also induced a larger reduction in levels of Arc, a downstream protein of the p18-mTORC1 pathway, in WT mice than in AS mice. Work from the Costa‐Mattioli lab has clearly indicated that a low concentration of rapamycin has no effect on LTP in WT mice, while a high concentration of rapamycin impairs LTP (Stoica et al., 2011). Thus, it is conceivable that p18 KD impairs LTP in hippocampus from WT mice, due to its over‐inhibition of mTORC1 activity. To further address this point, we investigated the effect of MHY1485, an mTOR activator, on TBS-induced LTP in hippocampal slices from p18 KD WT mice. The results showed that acute activation of mTORC1 could improve TBS‐elicited LTP in these slices, indicating that a tightly controlled p18 level is critical for normal p18‐mTORC1 signaling and synaptic plasticity. These data have been shown in Figure 7 and we have expanded our discussion on this topic in the revised manuscript.

2) Figure 2C. The amount of GFP in the shScrambled versus shP18 groups appears different. Is this due to different image exposure times or different transfection efficiency? This is problematic vis-à-vis the accuracy quantification of p18 in the treatment group.

The shP18 AAV we use showed very good and consistent knockdown efficiency with GFP as a reliable marker and most of the infected neurons showed similar levels of GFP, which is now better shown in higher magnification images. Therefore, high magnification images of neurons with comparable GFP expression are used in the new Figure 2C.

3) Figure 4H. The authors claim that in Ube3a-deficient neurons, excessive mTORC1 is localized to the lysosome resulting in enhanced activity. Is it possible that both mTORC1 and mTORC2 are being localized to the lysosome under these conditions? How can the authors distinguish between these complexes given their data?

Recent studies have reported that mTORC1 is localized to the lysosome while mTORC2 is localized to mitochondria‐associated ER membrane (see review from Betz and Hall, 2013). However, we investigated the location of mTORC1 and mTORC2 in our study. We performed double staining of Raptor, which is specific for mTORC1, and Rictor, which is specific for mTORC2, with LAMP2. Raptor, but not Rictor, was clearly co‐localized with LAMP2 in CA1 pyramidal neurons in both WT and AS mice, and more Raptor/LAMP2 double‐stained puncta were detected in AS mice than in WT mice, suggesting that mTORC1 but not mTORC2 is localized to the lysosome under these conditions. These new data have been included in the Figure 4—figure supplement 2B‐D.

4) Figure 5D. Treatment of cultured hippocampal neurons with shP18 in wild-type mice causes an increase in the number of spines but in Figure 7B, wild type mice treated with siP18 show a reduction in the number of mature spines. A rationale for this discrepancy should be included in the Discussion.

P18 knockdown increased total spine density in *both* cultured WT hippocampal neurons (Figure 5C‐D) and in vivo (Figure 8—figure supplement 1A). P18 KD also decreased the proportion of mature spines in vivo (Figure 8A‐B and Figure 8—figure supplement 1B). The proportion of mature spines does not always correlate with total spine density; p18 KD‐induced increase in total spine density in AS mice was associated with increased proportion and number of mature spines, while in WT mice, it was associated with increased number and density of immature spines. Here again, we postulate that differences in p18 levels and mTORC1/mTORC2 signaling are responsible for these differences. As discussed in the manuscript, UBE3A deficiency results in AS, while its overexpression increases risks for autism spectrum disorder. Whether spine maturation plays any roles in UBE3A over‐expression‐induced ASD is an interesting question to be addressed in future studies. Part of this answer has been incorporated in the revised manuscript.

5) Some of the immunofluorescent quantification data (Figure 2F and Figure 4B) are not accompanied by a representative image. These should be included with the corresponding figure.

The representative images for the quantification data in Figure 2F are in Figure 2C‐E, and for Figure 4B are in Figure 4A, and Figure 4—figure supplement 1A‐C. We have indicated this in the figure legends of the revised manuscript.

6) Some of the Western blots represent composite gels. Thus, the β-actin and GAPDH loading control are not valid.

We have included all full blots in the supplementary material.

7) Figure 1E and Figure 3C are missing input lanes.

The input lanes were shown in Figure 1—figure supplement 1A and Figure 3—figure supplement 1C respectively.

8) For some of the normalized data, error bars are missing for control groups.

Error bars have been added.

9) In Figure 3E, a counter stain such as DAPI should be included for visual confirmation of consistent exposure between groups.

As suggested, DAPI staining has been included in the images.

Reviewer #2:[…] 1) Figure 1—figure supplement 1A: It seems like p18 protein levels are dramatically reduced with His-ubiquitin expression, even in the absence of Ube3a expression. How do the authors interpret this?

There is endogenous Ube3a in COS‐1 cells, as shown in the first 3 lanes of the Ube3a blot in Figure 1—figure supplement 1A. We have clarified this in the revised manuscript. To avoid any misunderstanding, we have also replaced the p18 blot with a more representative one in the new figure.

2) Figure 1E and Figure 3C: y-axes are labeled as relative p18 ubiquitination. How is that calculated? Is it the ratio of ubiquitinated p18 over total p18?

Yes. It is the ratio of ubiquitinated p18 over total p18, and it was normalized to the average value of the control group. The description has been added in the Materials and methods.

3) Figure 3D: p18 protein levels are increase with MG132 treatment even in Ube3a KO mice. What is the explanation for this increase?

There is still a little expression of paternal Ube3a in the hippocampus of AS mice, as shown in Figure 3A, which has also been reported by several groups (Judson et al., 2014).

4) In the Results (subsection, “4. Increased p18 levels in AS mice are associated with increased lysosomal localization of the Ragulator-Rag complex and mTOR”), the authors study the localization of "p-mTOR, the active isoform of mTOR." I believe they are using Ser2448 phospho-mTOR antibody for that purpose. There is some controversy about the role of Ser2448 phosphorylation on mTOR activity. Substitution of Ser2448 with alanine does not affect mTOR activity (Sekulic et al., 2000). If there is good evidence that this phospho-specific antibody correlates with mTOR activity, it should be included in this section.

We agree with the reviewer that mTOR Ser2448 phosphorylation status does not necessarily predict mTOR activity, so we have modified the statement in the revised manuscript. In addition, we examined the co‐localization of Raptor with LAMP2. Raptor is a critical component of mTORC1 and serves as a scaffold to spatially position substrates in close proximity to mTOR (Hara et al., 2002; Kim et al., 2002; Nojima et al., 2003), the binding of which to Rag GTPases is necessary and sufficient to mediate amino acid signaling to mTORC1 (Sancak et al., 2008). The result showed that Raptor was clearly co‐localized with LAMP2 in CA1 pyramidal cell soma of adult mice, and more Raptor/LAMP2 double‐stained puncta were detected in AS than in WT mice (Figure 4—figure supplement 2B and D). Consistently, Western blot results showed that levels of mTOR and Raptor were markedly increased in lysosomal fractions of hippocampus from AS mice, as compared to WT mice (Figure 4—figure supplement 2E). These new data have been included in the revised manuscript.

5) S6 ribosomal protein can get phosphorylated at Ser235/236 by S6Ks; however, S6K1/2 are not the only enzymes that are able to phosphorylate these sites. The contribution of S6K1/2 in S6 phosphorylation was addressed using S6K1/S6K2 double knock-out animals, which were found to display no phosphorylation of S6 at Ser240/244, but persistent phosphorylation at Ser235/236 (Pende et al., Mol. Cell., 2004). Kinases other than S6K can phosphorylate Ser235/236. They should investigate whether phosphorylation of S6 at Ser240/244 is affected by p18.

As suggested, we examined the levels of p‐S6 at Ser240/244, and have included the new data in the revised manuscript (Figure 5A‐B and Figure 6—figure supplement 1D‐E).

6) Quantification method for both imaging and western blotting should be further described:

We have included the following description in the revised manuscript.

a) Specifically, for imaging quantification how was mean fluorescence intensity calculated, over the entire image field or a specific region of interest?

Mean fluorescence intensity (MFI) was calculated over a specific region of interest. Also, background staining of the sections was measured and subtracted from the total signal to obtain the specific signal.

b) I cannot figure out how Figure 2F was generated. I assume the authors combined quantification of different double-staining experiments. This should be clarified. It should also be clarified what the n refers to in the figure legend regarding Figure 2F.

Figure 2F is the combined quantification of double‐staining experiments shown in Figure 2C-E. N refers to the number of culture dishes analyzed.

c) The ideal figure to include in Figure 2 would be one where are representative images of p18 staining and LAMP staining simultaneously of neurons treated with p18 shRNA or control shRNA. I could not find such a figure in the manuscript.

As suggested, we have included p18/LAMP2/GFP staining images in the new Figure 2C.

7) The authors show that knockdown of p18 results in loss of lysosomal localization of the Ragulator complex and Rag in hippocampal neurons by immunofluorescence. This is an important finding. Another way to make this assertion more robust would be to prepare lysosomal fractions and demonstrate these findings biochemically.

As suggested, we isolated lysosomes from neuronal cultures treated with Accell control or p18 siRNA, and measured levels of the Ragulator‐Rag complex in the lysosomal fractions. Consistent with the immunofluorescence results, levels of p18, p14, LAMTOR4, as well as RagA and RagB were significantly reduced by p18 KD. The new data have been included in Figure 2G.

8) Figure 4: Do AS hippocampi have more lysosomes? How do we know that these lysosomes are neuronal vs. glia? There is no NeuN or other stain to determine the cell type that are being imaged.

According to the results of LAMP2 staining, AS hippocampi have about 20% more lysosomes than WT ones. As suggested, we performed double staining of NeuN and LAMP2 in hippocampal slices from WT and AS mice. The lysosomes are mostly in neurons, and the results have been included in the Figure 4—figure supplement 2A.

9) Figures 5B: Why is p-PKC/GAPDH ratio quantified? The quantification should be done showing the level of the phospho-protein relative to its respective total protein (phospho-protein/total protein) after their respective normalization to the house keeping protein.

Levels of both PKC and p‐PKC reflect mTORC2 activity, since previous reports indicated that mTORC2‐mediated PKCα phosphorylation was critical for its stability (Bornancin and Parker, 1997; Sarbassov et al., 2004). Phosphorylated and total levels of PKC therefore exhibit the same trend, and thus the ratio of p‐PKC over PKC would not reflect mTORC2 activity.

10) Figure 6—figure supplement 1 – only total PKC levels were reported. Is that meant to show phospho-PKC?

Phospho‐PKC is shown in Figure 6 and total PKC is shown in Figure 6—figure supplement 1. Also see reply above.

11) Along the same lines, Figure 5 p-mTOR and p-S6 are modest and probably the least convincing data of the paper. It would be good to include other direct targets substrates of mTOR as markers of mTOR activation (e.g. p70/S6K and 4EBP1).

As suggested, the data for p‐4EBP1/4EBP1 have been included in Figure 5A‐B.

12) A previous paper by Tang et al. (2014, Neuron, PMID: 25155956) argues that mTORC1 over-activation results in increased spines. How do the authors reconcile this report with their findings?

Tang et al. demonstrated that over‐activated mTORC1 in TSC2^+/‐^ ASD mice results in increased spines by inhibiting autophagy that underlies postnatal spine pruning. To date no report has indicated reduced autophagy in AS mice, this point has been discussed in the revised manuscript.

13) Discussion: The authors propose that over-expression of Arc may be mechanistic link between Ube3a deficiency and LTP deficit. This argument has some holes. In their previous publication for example, these authors showed that rapamycin treatment decreased Arc levels in WT hippocampal slices, but did not affect LTP or spine morphology in WT mice (Sun et al., 2016). It seems like it would be easy to check Arc protein levels in their system to see Arc is affected by p18 manipulations.

As suggested, we examined the levels of Arc in CA1 dendritic field of mice injected with control or p18 siRNA. The data are shown in Figure 7D‐E. As expected, levels of Arc were significantly higher in control siRNA‐injected AS mice, as compared to control siRNA‐injected WT mice; p18 KD significantly reduced Arc levels in both WT and AS hippocampal slices. Of note, p18 KD induced much lower expression of Arc in WT mice than in AS mice, while rapamycin treatment induced similar level of decrease in Arc expression in WT and AS mice in our previous report (Sun et al., 2016), which may provide an explanation for the different changes in LTP and spine morphology in WT mice. See also reply to the first question of reviewer #1. P18 silencing impairs LTP and dendritic spines in WT neurons due to its over‐inhibition of mTORC1 activity, and the resulting “too‐low” levels of Arc may play a role.

Reviewer #3:[…] Method.1) The manuscript is built on blots and IHC, please provide all source data for all experiments (blots) and add validation for Ab use (IHC). This is critical for any interpretation of the data. Assuming the authors provide the source data and control the first part of the manuscript is solid and important.

All source data for blots have been provided in the supplementary material, and validation for antibodies used in IHC has been described in the Materials and methods.

2) I did not see any description of the vectors? Which promoter they used? General neuronal specific? This can change the story accordingly (specific neuronal phenotype versus general).

We employ a dual convergent promoter system (U6 and H1 promoters) where the sense and antisense strands of the siRNA are expressed by two different promoters rather than in a hairpin loop ‐ to avoid any possible recombination events that can occur. Both U6 and H1 promoters are general promoters commonly used for driving shRNA/siRNA expression. The description of the vectors has been included in the revised manuscript.

The LTP phenotype is interesting however, the authors do not explain the rescue mechanistically, if its Arc dependent – show it, if there is an effect on protein synthesis – show it.

As suggested, we examined the levels of Arc in CA1 dendritic field injected with control or p18 siRNA. The data are shown in Figure 7D‐E. As expected, levels of Arc were significantly higher in control siRNA‐injected AS mice as compared to control siRNA‐injected WT mice; p18 KD significantly reduced Arc levels in both WT and AS hippocampal slices. Of note, p18 KD induced much lower expression of Arc in WT mice than in AS mice, which may contribute to reduced LTP in p18 siRNA WT group. Together, these results suggest that p18‐mediated regulation of Arc levels is critical for LTP in WT and AS mice.

In order to increase the manuscript impact, rescue behavioral phenotype using similar manipulation to CA1 region of the hippocampus is needed. The authors can use MWM and/or contextual fear conditioning.

As suggested, we tested fear conditioning in WT and AS mice injected with AAV vectors containing p18 siRNA or scrambled siRNA (control) in CA1 region of the hippocampus. AS mice were impaired in context‐dependent fear conditioning, and p18 downregulation in CA1 significantly enhanced context‐dependent learning performance in AS mice, while it impaired context‐dependent learning performance in WT mice, which is consistent with its effects on LTP and spine morphology. The data have been included in Figure 8E.

[Editors' note: the author responses to the re-review follow.]

The manuscript has been improved but there are some remaining issues that need to be addressed before acceptance, as outlined below:The revised manuscript by Sun et al. describes the novel interaction between Ube3a and p18 that results in regulation of neuronal mTORC1 signaling. The authors have addressed many of the critical questions and provided the source data to better evaluate the manuscript. In general, this study provides a novel mechanism of regulation of p18 by Ube3a ubiquitination. There is one major point below that should be addressed to strengthen the manuscript.The direct link the authors make between LTP and synaptic structure they measured is not that obvious. Kaphzan et al. reported changes in IPSP, but not EPASP in CA1 region in AS mice:Reversal of impaired hippocampal long-term potentiation and contextual fear memory deficits in Angelman syndrome model mice by ErbB inhibitors.Kaphzan H, Hernandez P, Jung JI, Cowansage KK, Deinhardt K, Chao MV, Abel T, Klann E.Biol Psychiatry. 2012 Mar 03;72(3):182-90. doi: 10.1016/j.biopsych.2012.01.021.Can the authors explain this discrepancy?

First, we want to point out that there is no agreement in the literature regarding whether there are changes in mEPSC and mIPSC in neurons lacking Ube3a. In contrast to what Kaphzan et al. (2012) reported, Greer et al. (2010) observed that, in both Ube3A RNAi-transfected hippocampal neurons and CA1 hippocampal neurons from AS mice, there was a decrease in mEPSC frequency but not amplitude, as compared to wild-type hippocampal neurons. Furthermore, they did not observe changes in either frequency or amplitude of mIPSCs from AS hippocampal slices, as compared to wild-type slices. Second, as Kaphzan et al. (2012) pointed out in their Discussion, “there are additional mechanisms by which ErbB inhibitors contribute to the rescue of LTP in AS mice” in addition to reducing inhibitory input to CA1 pyramidal neurons. We believe that there are a few not mutually exclusive mechanisms, which could impair LTP in AS mice. We have included these points of discussion in the fourth and sixth paragraphs of our revised Discussion.